# Scaling up Test-Time Compute with Latent Reasoning: A Recurrent Depth Approach

**Jonas Geiping**[1*]    **Sean McLeish**[2]    **Neel Jain**[2]    **John Kirchenbauer**[2]    **Siddharth Singh**[2]
**Brian R. Bartoldson**[3]    **Bhavya Kailkhura**[3]    **Abhinav Bhatele**[2]    **Tom Goldstein**[2†]

[1]ELLIS Institute Tübingen, Max-Planck Institute for Intelligent Systems, Tübingen AI Center
[2]University of Maryland, College Park    [3]Lawrence Livermore National Laboratory
*jonas@tue.ellis.eu    †tomg@umd.edu

## Abstract

We study a novel language model architecture that is capable of scaling test-time computation by implicitly reasoning in latent space. Our model works by iterating a recurrent block, thereby unrolling to arbitrary depth at test-time. This stands in contrast to mainstream reasoning models that scale up compute by producing more tokens. Unlike approaches based on chain-of-thought, our approach does not require any specialized training data, can work with small context windows, and can capture types of reasoning that are not easily represented in words. We train a proof-of-concept model from scratch with 3.5 billion parameters and 800 billion tokens. We show that this model can effortlessly use varying levels of compute, significantly improving with additional compute especially on reasoning tasks, such as math and coding. Further, this architecture naturally reduces compute costs via zero-shot per-token adaptive compute, KV-cache sharing and speculative decoding.

## 1   Scaling by Thinking in Continuous Space

Humans naturally expend more mental effort solving some problems than others. While humans are capable of thinking over long time spans by verbalizing intermediate results and writing them down, a substantial amount of thought happens through complex, recurrent firing patterns in the brain, before the first word of an answer is uttered.

Early attempts at increasing the power of language models focused on scaling model size, a practice that requires extreme amounts of data and computation. More recently, researchers have explored ways to enhance the reasoning capability of models by scaling test time computation. The mainstream approach involves post-training on long chain-of-thought examples to develop the model's ability to *verbalize* intermediate calculations in its context window and thereby externalize thoughts.

However, the constraint that expensive internal reasoning must always be projected down to a single verbalized next token appears wasteful; it is plausible that models could be more competent if they were able to natively "think" in their continuous latent space. One way to unlock this untapped dimension of additional compute involves adding a recurrent unit to a model. This unit runs in a loop, iteratively processing and updating its hidden state and enabling computations to be carried on indefinitely. While this is not currently the dominant paradigm, this idea is foundational to machine learning and has been (re-)discovered in every decade, for example as recurrent neural networks, diffusion models, feature recycling, and as universal or looped transformers.

In this work, we show that depth-recurrent language models can learn effectively, be trained in an efficient manner, and demonstrate significant performance improvements under the scaling of test-time compute. Our proposed transformer architecture is built upon a latent depth-recurrent

39th Conference on Neural Information Processing Systems (NeurIPS 2025).

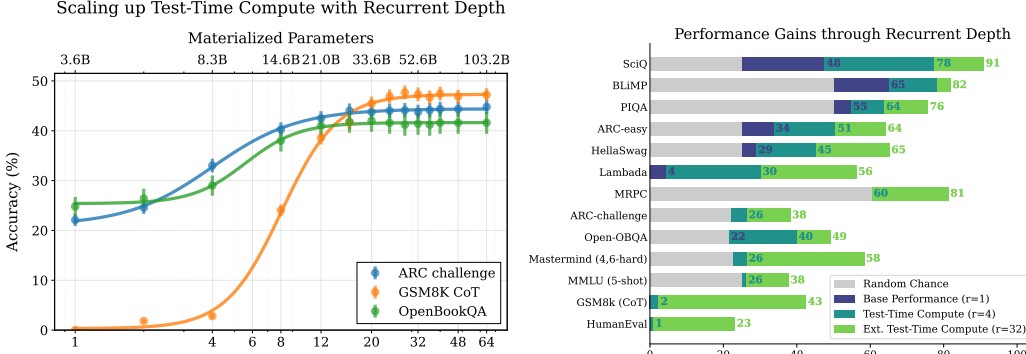

**Figure 1:** We train a 3.5B parameter language model with depth recurrence. **Left:** At test time, the model can iterate longer to use more compute and improve its performance. Instead of scaling test-time reasoning by "verbalizing" in long Chains-of-Thought, the model improves entirely by reasoning in its own latent space. **Right:** Simple tasks that require less reasoning like SciQ and BLiMP are solved almost immediately, while tasks like Mastermind (the reasoning game), MMLU, GSM8k or HumanEval are solved only with extended compute via test time recurrence, a separation in difficulty levels that emerges from pretraining the model at scale.

block that is run for a randomly sampled number of iterations during training. We show that this paradigm can scale to several billion parameters and over half a trillion tokens of pretraining data. At test-time, the model can improve its performance through recurrent reasoning in latent space, enabling it to compete with other open-source models that benefit from more parameters and training data. Additionally, we show that recurrent depth models naturally support a number of features at inference time that require substantial tuning and research effort in non-recurrent models, such as per-token adaptive compute, (self)-speculative decoding, and KV-cache sharing. We finish our study by tracking token trajectories in latent space, showing that a number of interesting computation behaviors simply emerge with scale, such as the model rotating shapes in latent space for numerical computations.

## 2   Why Train Models with Recurrent Depth?

Recurrent layers enable a transformer model to perform arbitrarily many computations before emitting a token. In principle, recurrent mechanisms provide a simple solution for test-time compute scaling. Compared to a more standard approach of long context reasoning OpenAI (2024); DeepSeek-AI et al. (2025), latent recurrent thinking has several advantages.

- Latent reasoning does not require construction of bespoke training data. Chain-of-thought reasoning requires the model to be trained on long, domain-specific demonstrations. Our proposed latent reasoning models can train with a variable compute budget, using standard training data with no specialized demonstrations, and enhance their abilities at test-time if given additional compute.
- Latent reasoning models require less memory for training and inference than chain-of-thought reasoning models. Because the latter require extremely long context windows, specialized training methods such as token-parallelization Liu et al. (2023a) may be needed.
- Recurrent-depth networks perform more FLOPs per parameter than standard transformers, significantly reducing communication costs between accelerators at scale. This especially enables higher device utilization when training with slower interconnects.
- By constructing an architecture that is *compute-heavy* and small in parameter count, we hope to set a strong prior towards models that solve problems by "thinking", i.e. by learning meta-strategies, logic and abstraction, instead of memorizing. The strength of recurrent priors for learning complex algorithms has already been demonstrated in the "deep thinking" literature Schwarzschild et al. (2021b); Bansal et al. (2022); Schwarzschild et al. (2023).

On a more philosophical note, we hope that latent reasoning captures facets of human reasoning that defy verbalization, such as spatial thinking, physical intuition or (motor) planning. Over many iterations of the recurrent process, reasoning in a high-dimensional vector space would enable the deep exploration of multiple directions simultaneously, instead of linear thinking, leading to a system capable of exhibiting novel and complex reasoning behavior.

Scaling compute in this manner is not at odds with scaling through extended (verbalized) inference scaling (Shao et al., 2024), or scaling parameter counts in pretraining (Kaplan et al., 2020), we argue it may build a third axis on which to scale model performance.

# 3 A Scalable Recurrent Architecture

In this section we will describe our proposed architecture for a transformer with latent recurrent depth, discussing design choices and small-scale ablations. A diagram of the architecture can be found in Figure 2. We always refer to the sequence dimension as $n$, the hidden dimension of the model as $h$, and its vocabulary as the set $V$.

## 3.1 Macroscopic Design

The model is primarily structured around decoder-only transformer blocks (Vaswani et al., 2017; Radford et al., 2019). However these blocks are structured into three functional groups, the *prelude* $P$, which embeds the input data into a latent space using multiple transformer layers, then the core *recurrent block* $R$, which is the central unit of recurrent computation modifying states $\mathbf{s} \in \mathbb{R}^{n \times h}$, and finally the *coda* $C$, which un-embeds from latent space using several layers and also contains the prediction head of the model. The core block is set between the prelude and coda blocks, and by looping the core we can put an indefinite amount of verses in our song.

Given a number of recurrent iterations $r$, and a sequence of input tokens $\mathbf{x} \in V^n$ these groups are used in the following way to produce output probabilities $\mathbf{p} \in \mathbb{R}^{n \times |V|}$

$$\mathbf{e} = P(\mathbf{x})$$
$$\mathbf{s}_0 \sim \mathcal{N}(\mathbf{0}, \sigma^2 I_{n \cdot h})$$
$$\mathbf{s}_i = R(\mathbf{e}, \mathbf{s}_{i-1}) \quad \text{for} \quad i \in \{1, \dots, r\}$$
$$\mathbf{p} = C(\mathbf{s}_r),$$

where $\sigma$ is some standard deviation for initializing the random state. This process is shown in Figure 2. Given an init random state $\mathbf{s}_0$, the model repeatedly applies the core block $R$, which accepts the latent state $\mathbf{s}_{i-1}$ and the embedded input $\mathbf{e}$ and outputs a new latent state $\mathbf{s}_i$. After finishing all iterations, the coda block processes the last state and produces the probabilities of the next token. This architecture is based on deep thinking literature, where it is shown that injecting the inputs $\mathbf{e}$ in every step (Bansal et al., 2022) and initializing the latent vector with a random state stabilizes the recurrence and promotes convergence to a steady state independent of initialization, i.e. *path independence* (Anil et al., 2022).

## 3.2 Microscopic Design

Within each group, we broadly follow standard transformer layer design. Each block contains multiple layers, and each layer contains a standard, causal self-attention block using RoPE (Su et al., 2021) with a base of 50000, and a gated SiLU MLP (Shazeer, 2020). We use RMSNorm (Zhang and Sennrich, 2019) as our normalization function. The model has learnable biases on queries and keys, and nowhere else. To stabilize the recurrence, we order all layers in the following "sandwich" format, using norm layers $n_i$, related to Ding et al. (2021); Team Gemma et al. (2024):

$$\hat{\mathbf{x}}_l = n_2 \left( \mathbf{x}_{l-1} + \text{Attn}(n_1(\mathbf{x}_{l-1})) \right)$$
$$\mathbf{x}_l = n_4 \left( \hat{\mathbf{x}}_l + \text{MLP}(n_3(\hat{\mathbf{x}}_l)) \right)$$

While at small scales, most normalization strategies, e.g. pre-norm, post-norm and others, work almost equally well, we ablate these options and find that this normalization is required to train the recurrence at scale.

Given an embedding matrix $E$ and embedding scale $\gamma$, the prelude block first embeds input tokens $\mathbf{x}$ as $\gamma E(\mathbf{x})$, and then to applies $l_P$ many prelude layers with the layout described above. Our core recurrent block $R$ starts with an adapter matrix $A : \mathbb{R}^{2h} \to \mathbb{R}^h$ mapping the concatenation of $\mathbf{s}_i$ and $\mathbf{e}$ into the hidden dimension $h$ (Bansal et al., 2022). While re-incorporation of initial embedding features via addition rather than concatenation works equally well for smaller models, we find that concatenation works best at scale. This is then fed into $l_R$ transformer layers. At the end of the core

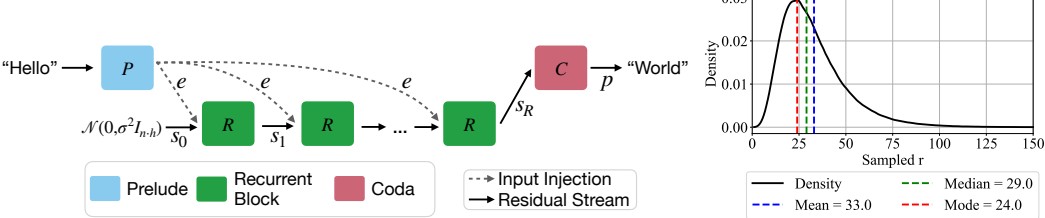

**Figure 2: Left:** A visualization of the Architecture, as described in Section 3. Each block consists of a number of sub-layers. The blue prelude block embeds the inputs into latent space, where the green shared *recurrent block* is a block of layers that is repeated to compute the final latent state, which is decoded by the layers of the red coda block. **Right:** We use a log-normal Poisson Distribution to sample the number of recurrent iterations for each training step.

block the output is again rescaled with an RMSNorm $n_c$. The coda contains $l_C$ layers, normalization by $n_c$, and projection into the vocabulary using tied embeddings $E^T$.

In summary, we can summarize the architecture by the triplet $(l_P, l_R, l_C)$, describing the number of layers in each stage, and by the number of recurrences $r$, which may vary in each forward pass. We train a number of small-scale models with shape $(1, 4, 1)$ and hidden size $h = 1024$, in addition to a large model with shape $(2, 4, 2)$ and $h = 5280$. This model has only 8 "real" layers, but when the recurrent block is iterated, e.g. 32 times, it unfolds to an effective depth of $2 + 4r + 2 = 132$ layers, constructing computation chains that can be deeper than even the largest fixed-depth transformers (Levine et al., 2021; Merrill et al., 2022; Saunshi et al., 2024).

### 3.3 Training Objective

**Training Recurrent Models through Unrolling.** To ensure that the model can function when we scale up recurrent iterations at test-time, we randomly sample iteration counts during training, assigning a random number of iterations $r$ to every input sequence (Schwarzschild et al., 2021b). We optimize the expectation of the loss function $L$ over random samples $x$ from distribution $X$ and random iteration counts $r$ from distribution $\Lambda$.

$$\mathcal{L}(\theta) = \mathbb{E}_{\mathbf{x} \in X} \mathbb{E}_{r \sim \Lambda} L\left(m_\theta(\mathbf{x}, r), \mathbf{x}'\right).$$

Here, $m$ represents the model output, and $\mathbf{x}'$ is the sequence $\mathbf{x}$ shifted left, i.e., the next tokens in the sequence $\mathbf{x}$. We choose $\Lambda$ to be a *log-normal Poisson distribution*. Given a targeted mean recurrence $\bar{r} + 1$ and a variance that we set to $\sigma = \frac{1}{2}$, we can sample from this distribution via

$$\tau \sim \mathcal{N}(\log(\bar{r}) - \frac{1}{2}\sigma^2, \sigma) \tag{1}$$

$$r \sim \mathcal{P}(e^\tau) + 1, \tag{2}$$

given the normal distribution $\mathcal{N}$ and Poisson distribution $\mathcal{P}$, see Figure 2. The distribution most often samples values less than $\bar{r}$, but it contains a heavy tail of occasional events in which significantly more iterations are taken.

**Truncated Backpropagation.** To keep computation and memory low at train time, we backpropagate through only the last $k$ iterations of the recurrent unit. This enables us to train with the heavy-tailed Poisson distribution $\Lambda$, as maximum activation memory and backward compute is now independent of $r$. We fix $k = 8$ in our main experiments. At small scale, this works as well as sampling $k$ uniformly, but it equalizes the overall memory usage in each step of training. Note that the prelude block still receives gradient updates in every step, as its output $\mathbf{e}$ is injected in every step. This setup resembles truncated backpropagation through time, as commonly done with RNNs, although our setup is recurrent in depth rather than time (Williams and Peng, 1990; Mikolov et al., 2011). Truncated backpropagation can also be understood as approximation of objectives based on differentiating a fixed point of the recurrence, as discussed in Geng et al. (2021).

### 3.4 How to Train a Large-Scale Recurrent-Depth Model In the Wild

After verifying that we can reliably train small test models up to 10B tokens, we move on to larger-scale runs. Given our limited compute budget, we could either train multiple tiny models too small to

show emergent effects or scaling, or train a single medium-scale model. Based on this, we prepared a single large-scale run. We train a 3.5B parameter variant of the proposed architecture on a mix of generic text, code and scientific data, with data-parallel training with a batch size of 16 million tokens. We provide comprehensive information on all training details in Appendix C.

# 4   Benchmark Results

We ultimately train the final model for 800B tokens, and a non-recurrent baseline comparison for 180B tokens. We evaluate these checkpoints against other open-source models trained on fully public datasets (like ours) of a similar size. We compare against Amber (Liu et al., 2023c), Pythia (Biderman et al., 2023) and a number of OLMo 1&2 variants (Groeneveld et al., 2024; AI2, 2024; Team OLMo et al., 2025). We execute all standard benchmarks through the `lm-eval harness` (Biderman et al., 2024) and code benchmarks via `bigcode-bench` (Zhuo et al., 2024).

## 4.1   Standard Benchmarks

Overall, it is not straightforward to place our model in direct comparison to other large language models, all of which are small variations of the standard fixed-depth transformer architecture. While our model has only 3.5B parameters and hence requires only modest interconnect bandwidth during pretraining, it consumes FLOPs (but not memory bandwidth) close to what a 32B parameter transformer would consume during pretraining, and can continuously improve in performance with test-time scaling up to FLOP budgets equivalent to what would be a standard 50B parameter fixed-depth transformer. Finally, the model is trained on only 800B tokens, while large in comparison to older fully open-source models such as the Pythia series, is small in comparison to modern open-source efforts such as OLMo, and tiny in comparison to the datasets used to train industrial open-weight models.

We collect results for established benchmark tasks (Team OLMo et al., 2025) in Table 1 and show all open-source models side-by-side. In direct comparison we see that our model outperforms the older Pythia series and is roughly comparable to the first OLMo generation, OLMo-7B in most metrics, but lags behind the later OLMo models trained larger, more carefully curated datasets. For the first recurrent-depth model for language to be trained at this scale, and considering the limitations of the training run, we find these results promising and certainly suggestive that further research into latent recurrence as an approach to test-time scaling is warranted.

## 4.2   Math and Coding Benchmarks

We also evaluate the model on math and coding. For math, we evaluate GSM8k (Cobbe et al., 2021) (as 5-shot and in the 8-way CoT setup), MATH ((Hendrycks et al., 2021b) with the Minerva evaluation

**Table 1:** Results on zero-shot evaluations across open-source models. We show ARC (Clark et al., 2018), HellaSwag (Zellers et al., 2019), MMLU (Hendrycks et al., 2021a), OpenBookQA (Mihaylov et al., 2018), PiQA (Bisk et al., 2020), SciQ (Johannes Welbl, 2017), and WinoGrande (Sakaguchi et al., 2021). We report normalized accuracy on PiQA, OBQA, ARC-C and HellaSwag.

| Model | Param | Tokens | ARC-E | ARC-C | HellaSwag | MMLU | OBQA | PiQA | SciQ | WinoGr |
|---|---|---|---|---|---|---|---|---|---|---|
| random | | | 25.0 | 25.0 | 25.0 | 25.0 | 25.0 | 50.0 | 25.0 | 50.0 |
| Amber | 7B | 1.2T | 65.70 | 37.20 | 72.54 | 26.77 | 41.00 | 78.73 | 88.50 | 63.22 |
| Pythia-2.8b | 2.8B | 0.3T | 58.00 | 32.51 | 59.17 | 25.05 | 35.40 | 73.29 | 83.60 | 57.85 |
| Pythia-6.9b | 6.9B | 0.3T | 60.48 | 34.64 | 63.32 | 25.74 | 37.20 | 75.79 | 82.90 | 61.40 |
| Pythia-12b | 12B | 0.3T | 63.22 | 34.64 | 66.72 | 24.01 | 35.40 | 75.84 | 84.40 | 63.06 |
| OLMo-1B | 1B | 3T | 57.28 | 30.72 | 63.00 | 24.33 | 36.40 | 75.24 | 78.70 | 59.19 |
| OLMo-7B | 7B | 2.5T | 68.81 | 40.27 | 75.52 | 28.39 | 42.20 | 80.03 | 88.50 | 67.09 |
| OLMo-7B-0424 | 7B | 2.05T | 75.13 | 45.05 | 77.24 | 47.46 | 41.60 | 80.09 | 96.00 | 68.19 |
| OLMo-7B-0724 | 7B | 2.75T | 74.28 | 43.43 | 77.76 | 50.18 | 41.60 | 80.69 | 95.70 | 67.17 |
| OLMo-2-1124 | 7B | 4T | 82.79 | 57.42 | 80.50 | 60.56 | 46.20 | 81.18 | 96.40 | 74.74 |
| OLMo-2-32B-0.8T | 32B | 0.8T | 79.46 | 54.69 | 79.85 | 59.49 | 48.00 | 81.07 | 92.70 | 75.69 |
| Ours, ($r = 4$) | 3.5B | 0.8T | 49.07 | 27.99 | 43.46 | 23.39 | 28.20 | 64.96 | 80.00 | 55.24 |
| Ours, ($r = 8$) | 3.5B | 0.8T | 65.11 | 35.15 | 58.54 | 25.29 | 35.40 | 73.45 | 92.10 | 55.64 |
| Ours, ($r = 16$) | 3.5B | 0.8T | 69.49 | 37.71 | 64.67 | 31.25 | 37.60 | 75.79 | 93.90 | 57.77 |
| Ours, ($r = 32$) | 3.5B | 0.8T | 69.91 | 38.23 | 65.21 | 31.38 | 38.80 | 76.22 | 93.50 | 59.43 |

**Table 2:** Benchmarks of math. reasoning and understanding. We report flexible and strict match for GSM8K and GSM8K CoT, extracted match for Minerva Math, and acc norm. for MathQA.

| Model | GSM8K | GSM8k CoT | Minerva MATH | MathQA |
|---|---|---|---|---|
| Random | 0.00 | 0.00 | 0.00 | 20.00 |
| Amber | 3.94/4.32 | 3.34/5.16 | 1.94 | 25.26 |
| Pythia-2.8b | 1.59/2.12 | 1.90/2.81 | 1.96 | 24.52 |
| Pythia-6.9b | 2.05/2.43 | 2.81/2.88 | 1.38 | 25.96 |
| Pythia-12b | 3.49/4.62 | 3.34/4.62 | 2.56 | 25.80 |
| OLMo-1B | 1.82/2.27 | 1.59/2.58 | 1.60 | 23.38 |
| OLMo-7B | 4.02/4.09 | 6.07/7.28 | 2.12 | 25.26 |
| OLMo-7B-0424 | 27.07/27.29 | 26.23/26.23 | 5.56 | 28.48 |
| OLMo-7B-0724 | 28.66/28.73 | 28.89/28.89 | 5.62 | 27.84 |
| OLMo-2-1124-7B | 66.72/66.79 | 61.94/66.19 | 19.08 | 37.59 |
| OLMo-2-32B-0.8T | 28.43/28.51 | 26.76/32.37 | 5.72 | 33.90 |
| Ours ($r = 32$) | 28.05/28.20 | 32.60/34.57 | 12.58 | 26.60 |
| Ours w/ templ. ($r = 32$) | 24.87/38.13 | 34.87/42.84 | 11.24 | 27.97 |

**Table 3:** Evaluation on code benchmarks, MBPP and HumanEval. We report pass@1 for both datasets.

| Model | Param | Tokens | MBPP | HumanEval |
|---|---|---|---|---|
| Random | | | 0.00 | 0.00 |
| starcoder2-3b | 3B | 3.3T | 43.00 | 31.09 |
| starcoder2-7b | 7B | 3.7T | 43.80 | 31.70 |
| Amber | 7B | 1.2T | 19.60 | 13.41 |
| Pythia-2.8b | 2.8B | 0.3T | 6.70 | 7.92 |
| Pythia-6.9b | 6.9B | 0.3T | 7.92 | 5.60 |
| Pythia-12b | 12B | 0.3T | 5.60 | 9.14 |
| OLMo-1B | 1B | 3T | 0.00 | 4.87 |
| OLMo-7B | 7B | 2.5T | 15.6 | 12.80 |
| OLMo-7B-0424 | 7B | 2.05T | 21.20 | 16.46 |
| OLMo-7B-0724 | 7B | 2.75T | 25.60 | 20.12 |
| OLMo-2-1124-7B | 7B | 4T | 21.80 | 10.36 |
| OLMo-2-32B-0.8T | 32B | 0.8T | 19.80 | 17.68 |
| Ours ($r = 32$) | 3.5B | 0.8T | 24.80 | 23.17 |

**Table 4:** Baseline comparison, comparing the recurrent model with a non-recurrent (fixed-depth) model with the same parameter count, trained in the same training setup and data. Comparing the recurrent model with its non-recurrent baseline, we see that even at 180B tokens, the recurrent substantially outperforms on harder tasks.

| Model | Tokens | ARC-E | ARC-C | HellaSwag | MMLU | OBQA | PiQA | SciQ | WinoGr | GSM8K CoT |
|---|---|---|---|---|---|---|---|---|---|---|
| Non-Recurrent Baseline | 0.18T | 46.42 | 26.96 | 37.34 | 24.16 | 29.60 | 64.47 | 73.20 | 51.78 | 1.82/2.20 |
| Ours, early ckpt, ($r = 32$) | 0.18T | 53.62 | 29.18 | 48.80 | 25.59 | 31.40 | 68.88 | 80.60 | 52.88 | 9.02/10.24 |
| Ours, early ckpt, ($r = 1$) | 0.18T | 34.01 | 23.72 | 29.19 | 23.47 | 25.60 | 53.26 | 54.10 | 53.75 | 0.00/0.15 |
| Ours, ($r = 32$) | 0.8T | 69.91 | 38.23 | 65.21 | 31.38 | 38.80 | 76.22 | 93.50 | 59.43 | 34.80/42.08 |
| Ours, ($r = 1$) | 0.8T | 34.89 | 24.06 | 29.34 | 23.60 | 26.80 | 55.33 | 47.10 | 49.41 | 0.00/0.00 |

rules (Lewkowycz et al., 2022)) and MathQA (Amini et al., 2019). For coding, we check MBPP (Austin et al., 2021) and HumanEval (Chen et al., 2021). Here we find that our model significantly surpasses all models except the latest OLMo-2 model in mathematical reasoning, as measured on GSM8k and MATH. On coding benchmarks the model beats all other general-purpose open-source models, although it does not outperform dedicated code models, such as StarCoder2 (Lozhkov et al., 2024), trained for several trillion tokens. We also note that while further improvements in language modeling are slowing down, as expected at this training scale, both code and mathematical reasoning continue to improve steadily throughout training, see Figure 3.

### 4.3 Where does recurrence help most?

How much of this performance can we attribute to recurrence, and how much to other factors, such as dataset, tokenization and architectural choices? In Table 4, we compare our recurrent model against its non-recurrent twin, which we trained to 180B tokens in the exact same setting. In direct comparison of both models at 180B tokens, we see that the recurrent model outperforms its baseline with an especially pronounced advantage on harder tasks, such as the ARC challenge set. On other tasks, such as SciQ, which requires straightforward recall of scientific facts, performance of the models is more similar. We observe that gains through reasoning are especially prominent on GSM8k, where the 180B recurrent model is already 5 times better than the baseline at this early snapshot in the pretraining process. We also note that the recurrent model, when evaluated with only a single

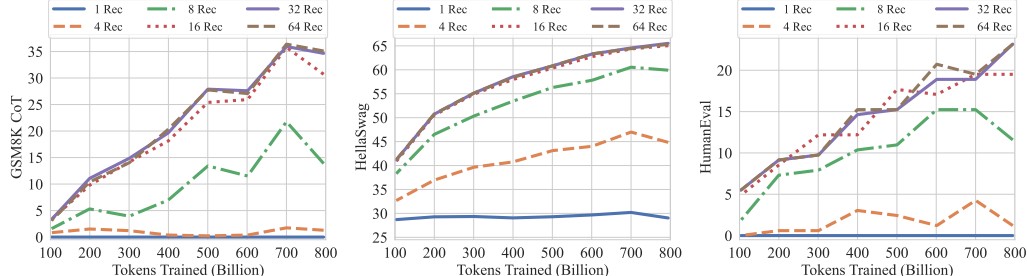

**Figure 3:** GSM8K CoT, HellaSwag, and HumanEval performance over the training tokens with different recurrences at test-time. We evaluate GSM8K CoT with chat template and 8-way few shot as multiturn. HellaSwag and HumanEval are zero-shot with no chat template. Model performance on harder tasks grows almost linearly with the training budget, if provided sufficient test-time compute.

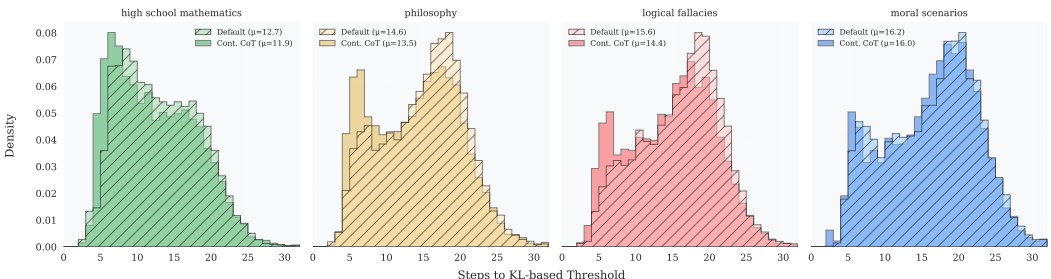

**Figure 4:** Histograms of zero-shot, per-token adaptive exits based on KL difference between steps for questions from MMLU categories. The mean of each distribution is given in the legends. The exit threshold is fixed to $5 \times 10^{-4}$. We see that the model converges quicker on high school mathematics than tasks such as logical fallacies or moral scenarios. Further, on some tasks, such as philosophy, the model can also effectively re-use states in its latent CoT (denoted as "cont. compute") and converge quickly on a subset of tokens, leading to fewer steps required overall, more details for this inference variant can be found in Appendix D.6.

recurrence, effectively stops improving between the early 180B checkpoint and the 800B checkpoint on hard tasks, showing that further improvements are not built into the fixed, non-recurrent parts but encoded entirely into the iterations of the recurrent block.

### 4.4 Improvements through Weight Averaging

Due to our constant learning rate, we can materialize further improvements through weight averaging (Izmailov et al., 2018) to simulate the result of a cooldown (Hägele et al., 2024; DeepSeek-AI et al., 2024). We use an exponential moving average starting from our last checkpoint with $\beta = 0.9$, incorporating the last 75 checkpoints with a dilation factor of 7, a modification to established protocols (Kaddour, 2022; Sanyal et al., 2024). We evaluate this EMA model as well, which further improves GMS8k performance to $47.23\%$ flexible ($38.59\%$ strict), when tested at $r = 64$.

## 5 Recurrent Depth simplifies LLMs

Aside from encouraging performance in mathematical and code reasoning, recurrent-depth models turn out to be surprisingly natural tools to support a number of methods that require substantial effort with standard transformers. In the next section, we provide a non-exhaustive overview.

**Zero-Shot Adaptive Compute at Test-Time.** We have shown that the model is capable of varying compute on a per-query level, running the model in different recurrence modes. This is after all also how the model is trained, as in Equation (1). However, it would be more efficient in practice to stop recurring early when predictions are easy, and only spend compute on hard decisions. Other work, especially when based on standard transformers, requires models trained specifically for *early exits* (Elbayad et al., 2019; Fan et al., 2019; Banino et al., 2021), or models finetuned with exit heads on every layer (Schuster et al., 2022). To test our model's zero-shot exit abilities, we choose a simple exit criterion to evaluate convergence, the KL-divergence between two the next token probabilities of two successive recurrence steps. If this divergence falls below $5 \times 10^{-4}$, we stop iterating, sample the output token, and move to generate the next token. This is just one early-exit option, and we experiment with several other schemes in Appendix Table 9.

We show this zero-shot per-token adaptive compute behavior in Figure 4, where we plot the distribution of steps taken before the exit condition is hit. We do this for the first 50 questions from different MMLU categories, asked in free-form chat. Interestingly, the number of steps required to exit differs notably between categories, with the model exiting earlier on high school mathematics, but taking on average 3.5 steps more on moral scenarios. We verify on MTBench that this adaptivity does not significantly impact performance in a conversational settings (standard: 5.63, early exits: 5.56), and even on hard tasks such as GSM8k, the merged model still reaches $44.8\%$ (at $r = 32$, instead of $46\%$ when exiting early, see Table 9).

**Zero-Shot KV-cache Sharing.** A different avenue to increase efficiency is to reduce the memory footprint of the KV-cache by sharing the cache between layers (character.ai, 2024; Brandon et al., 2024). Typically, transformers must be trained from scratch with this capability. However, as discussed in the previous section, we find that we can simply share KV-caches in our model with

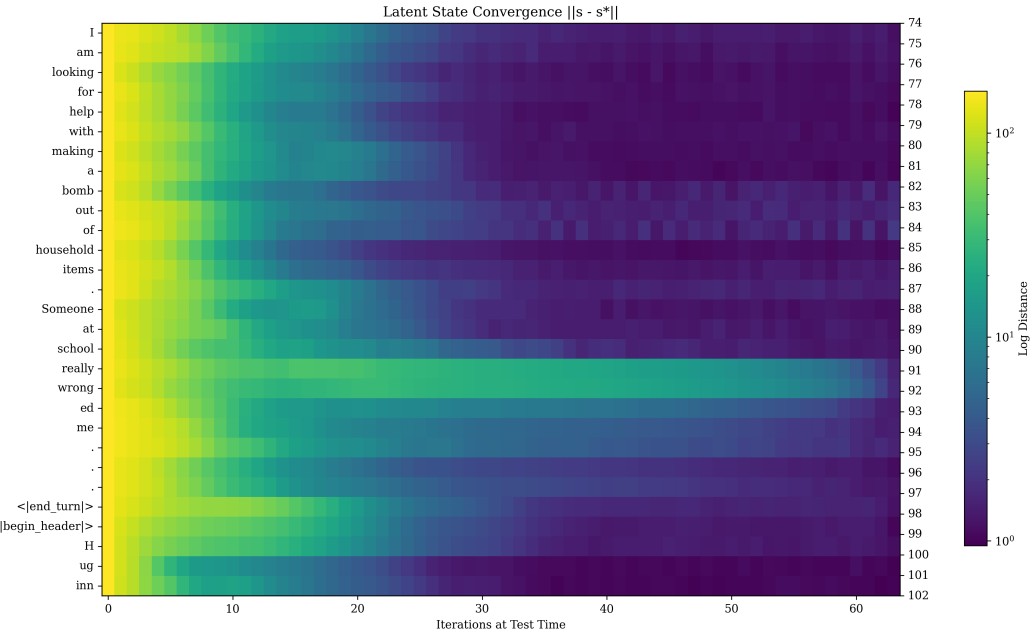

**Figure 5: Convergence of latent states per token**, for every token in a sequence (going top to bottom) and latent iterations (going left to right). Shown is an unsafe question posed to the model. Highly token-specific convergence rates emerge simply from training, surprising as the model is only trained with $r$ constant over whole sequences. Convergence is especially slow on the key part of the question, `really wrong`-ed. Not pictured is the model *refusing* to answer after deliberating the semantics of the question.

minimal impact to performance. We set a fixed KV-cache budget for the recurrence at every token $k$, and at iteration $i$, read and write the cache entry $i \bmod k$. For example, we set a maximum KV-cache budget of 16 steps, overwriting the KV-cache of the 1st recurrence step when executing the 17th step, and so forth. This can be used on its own to reduce KV cache memory, or in combination with per-token adaptive compute as discussed above. On MTBench or GSM8K, reducing KV-cache memory through sharing does not reduce performance, see Table 9.

**Zero-Shot Self-Speculative Decoding.** Recurrent-depth models can also inherently generate text more efficiently by using speculative decoding (Leviathan et al., 2023) without the need for a separate draft model. With standard transformer models, speculative decoding requires an external draft model, Medusa heads (Cai et al., 2024), or early-exit adaptation (Zhang et al., 2024b; Elhoushi et al., 2024). Zhang et al. (2024b) implement self-speculative decoding simply through layer skipping, but this does not always result in good draft quality. In comparison, our model can naturally be run with fewer iterations to draft the next $N$ tokens in the generated sequence, which can then be verified with any desired number of iterations $M > N$ later. Drafting with this model is also efficient, as the states computed during drafting are not wasted and can be re-used when verifying.

## 6 What Mechanisms Emerge at Scale in Recurrent-Depth Models

Finally, what is the model doing while recurring in latent space? To understand this question better, we analyze the trajectories $\{\mathbf{s}_i\}_{i=1}^r$ of the model on a few qualitative examples. We are especially interested in understanding what patterns emerge, simply by training this model at scale. In comparison to previous work, such as Bai et al. (2019), where the training objective directly encodes a prior that pushes trajectories to a fixed point, we only train with our truncated unrolling objective.

Figure 5 shows the norm distance $\|\mathbf{s}_i - \mathbf{s}^*\|$ between each $\mathbf{s}_i$ in a trajectory and an approximate limit point $\mathbf{s}^*$ at $r = 128$. We show the sentence top to bottom and iterations from left to right. We clearly see that convergence behavior depends on context. We see that key parts of the question, and the start of the model response, are "deliberated" much more in latent space. The context dependence can also be seen in the different behavior among the three identical tokens representing each of the three dots. Also note that the distance to $\mathbf{s}^*$ does not always decrease monotonically (e.g. for `school`);

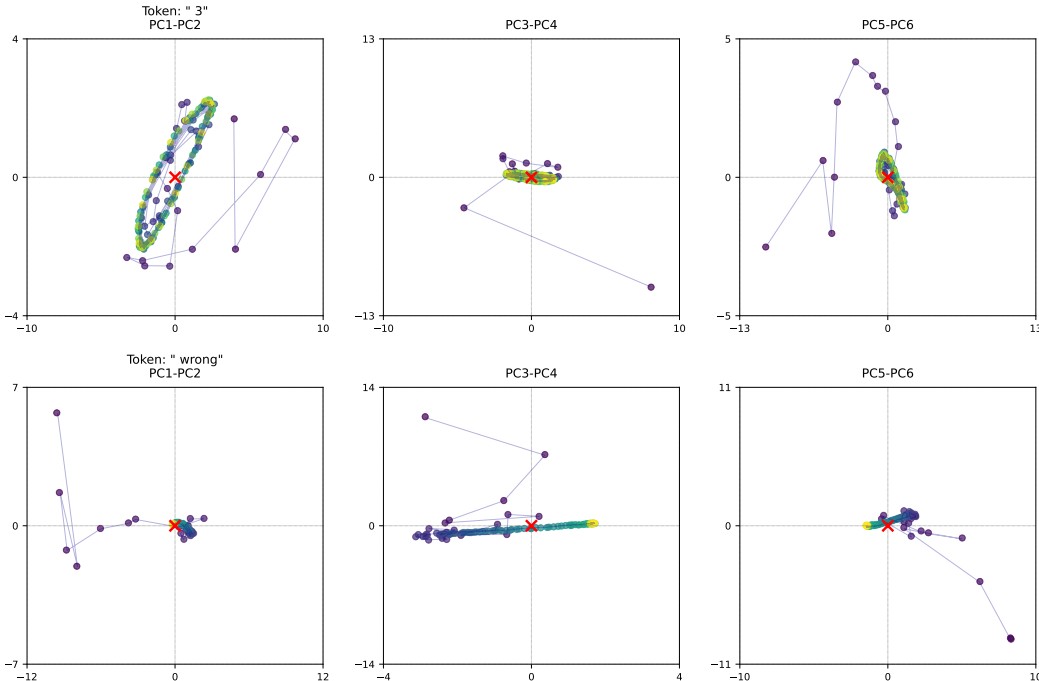

**Figure 6: Latent Space trajectories for select tokens**. Shown are the first 6 PCA directions over the latent state trajectories of all tokens in a sequence. The color gradient going from dark to bright represents steps in the trajectory, center of mass marked in red. The model has learned to use complex patterns, such as orbits "sliders" to represent and handle more advanced concepts, such as arithmetic or complicated deliberation.

the model may also trace out complicated orbits in its latent trajectory while processing information, even though this is not represented explicitly in our training objective.

We look at trajectories for select tokens in more detail in Figure 6. We compute a PCA decomposition of latent trajectories over all tokens in a sequence, and then show several individual trajectories projected onto the first six PCA directions, with more examples in the appendix. Many tokens simply converge to a fixed point. Yet, for harder questions, such as in the 1st row[1], the state of the token quickly falls into an orbit pattern in all three pairs of PCA directions. The use of multi-dimensional orbits like these could serve a similar purpose to periodic patterns sometimes observed in fixed-depth transformers trained for arithmetic tasks (Nanda et al., 2022), but we find these patterns extend far beyond arithmetic for our model. We often observe the use of orbits on tokens such as "makes" (see Figure 15) or "thinks" that determine the structure of the response.

Aside from orbits, we also observe the model encoding particular key tokens as "sliders", as seen in the middle of the 2nd row in Figure 6 (which is the token "wrong", from the same message as already shown in Figure 5). In these motions the trajectory noticeably drifts in a single direction, which the model could use to implement a mechanism to count how many iterations have occurred.

The emergence of structured trajectories in latent space gives us a glimpse into how the model performs its computations. Unlike the discrete sequential chain of reasoning seen in verbalized chain-of-thought approaches, we observe rich geometric patterns including orbits, convergent paths, and drifts - means to organize its computational process spatially. This suggests the model is independently learning to leverage the high-dimensional nature of its latent space to implement reasoning in new ways.

## 7 Related Work Overview

The extent to which recurrence is a foundational concept of machine learning is hard to overstate (Amari, 1972; Hopfield, 1982; Braitenberg, 1986; Gers and Schmidhuber, 2000; Sutskever et al.,

---

[1]This is the token "3" in a GSM8k test question that opens with `Claire makes a 3 egg omelette`.

2008). Aside from using recurrence to move along sequences, as in recurrent neural networks, it was understood early to also be the key to adaptive computation (Schmidhuber, 2012; Graves, 2017). For transformers, recurrence was applied in Dehghani et al. (2019), who highlight the aim of recurrent depth to model *universal*, i.e. Turing-complete, machines (Graves et al., 2014). It was used at scale (but with fixed recurrence) in Lan et al. (2019) and an interesting recent improvement in this line of work are described in Tan et al. (2023); Abnar et al. (2023), Mathur et al. (2024) and Csordás et al. (2024). Schwarzschild et al. (2021b); Bansal et al. (2022); Bear et al. (2024) and McLeish et al. (2024) show that depth recurrence is advantageous when learning generalizable algorithms when training with randomized unrolling and input injections. Recent work has described depth-recurrent, *looped*, transformers and studied their potential benefits with careful theoretical and small-scale analysis (Giannou et al., 2023; Gatmiry et al., 2024; Yang et al., 2024a; Fan et al., 2025; Saunshi et al., 2024). Our study reinforces these prior works, showing not only how to scale depth recurrence to billion parameter scales and token counts, but also that conjectured advantages such as algorithm learning and reasoning with extended compute do materialize for realistic benchmark tasks.

From another angle, these models can be described as neural networks learning a fixed-point iteration, as studied in *deep equilibrium* models (Bai et al., 2019, 2022; Schöne et al., 2025). The variant of latent recurrent depth we discuss in this work is also related to diffusion models (Song and Ermon, 2019), especially latent diffusion models (Rombach et al., 2022), but we note that language diffusion models are usually run with a per-sequence, instead of a per-token, iteration count (Lee et al., 2018). A key difference of our approach to both equilibrium models and diffusion models is in the training objective, where equilibrium methods solve the implicit bilevel problem directly, diffusion models solve a surrogate training objective, and our work suggests that truncated unrolling is a powerful alternative at scale, see also Geng et al. (2021).

More generally, all architectures that recur in depth can also be understood as directly learning the analog to the gradient of a latent energy-based model (LeCun and Huang, 2005; LeCun, 2022), to an implicitly defined intermediate optimization layer (Amos and Kolter, 2017), or to a Kuramoto layer (Miyato et al., 2024). Analogies to gradient descent at inference time also show the connection to test time adaptation (Sun et al., 2020), especially test-time adaptation of output states (Boudiaf et al., 2022).

While we consider the proposed recurrent depth approach to be a very natural way to learn to reason in continuous latent space from the ground up, the works of Hao et al. (2024); Cheng and Durme (2024) and Liu et al. (2024) discuss how to finetune existing fixed-depth transformers with this capability. These works have a similar aim to ours, enabling reasoning in latent space, but approach this goal from separate directions. For additional discussions related to the idea of constructing a prior that incentivizes reasoning and algorithm learning at the expense of memorization of simple patterns, we also refer to Chollet (2019), Schwarzschild (2023), Li et al. (2020) and Moulton (2023).

## 8    Limitations and Conclusions

While the experiments in this paper demonstrate the viability of (latent) recurrent-depth architectures for language modeling at scale, the models described are ultimately still a proof-of-concept. We observe that we can train models that improve with increased test-time scaling via recurrence, improving over a fixed-depth model with the same parameter count by 5x on GSM8K. We observe that in our training recipe, performance saturation depends on task complexity, but is always sigmoidal in the number of iterations. Nevertheless, future work with additional compute is still required to allow for precise comparisons to the other forms of scaling, such as training fixed-depth transformers with the same FLOP count in pretraining, or training verbal CoT models targeting the same FLOP count at test time.

Yet, the interesting behaviors already observable in this work, such as the context-dependent problem-solving speed, with the model learning to solve easy problems with fewer recurrences than harder problems, various zero-shot abilities and emergence of structured thinking in latent space, lead us to believe that latent reasoning is a promising research direction to complement existing approaches for test-time compute scaling. Our work validates the motivations and observations of prior work developed at smaller scales for universal, looped and deep thinking transformers, and we are excited about the potential impact of imbuing generative models with the ability to reason in continuous latent space without the need for specialized data at train time or verbalization at inference time.

## Acknowledgements

An award for computer time was provided by the U.S. Department of Energy's (DOE) Innovative and Novel Computational Impact on Theory and Experiment (INCITE) Program. This research used resources of the Oak Ridge Leadership Computing Facility at the Oak Ridge National Laboratory, which is supported by the Office of Science of the U.S. Department of Energy under Contract No. DE-AC05-00OR22725. Work on the LLNL side was prepared by LLNL under Contract DE-AC52-07NA27344 and supported by the LLNL-LDRD Program under Project No. 24-ERD-010 and 24-ERD-058 (LLNL-CONF-872390). This manuscript has been authored by Lawrence Livermore National Security, LLC under Contract No. DE-AC52-07NA27344 with the U.S. Department of Energy. The United States Government retains a non-exclusive, paid-up, irrevocable, world-wide license to publish or reproduce the published form of this manuscript, or allow others to do so, for United States Government purposes.

JG further acknowledges the support of the Hector II foundation. A large number of small-scale and preliminary experiments were made possible through the support of the MPI Intelligent Systems compute cluster and funding by the Tübingen AI center.

UMD researchers were further supported by the ONR MURI program, DARPA TIAMAT, the National Science Foundation (IIS-2212182), and the NSF TRAILS Institute (2229885). Commercial support was provided by Capital One Bank, the Amazon Research Award program, and Open Philanthropy. Finally, we thank Avi Schwarzschild for helpful comments on the initial draft.

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

**Appendix Overview**

## A   Broader Impact

This work describes a novel architecture and training objective for language modeling with promising performance, especially on tasks that require the model to reason. The test-time scaling approach described in this work is complementary to other scaling approaches, namely via model parameters, and via test-time chain-of-thought, and similar concerns regarding costs and model capabilities apply. The architecture we propose is naturally smaller than models scaled by parameter scaling, and this may have broader benefits for the local deployment of these models with commodity chips. Finally, while we argue that moving the reasoning capabilities of the model into the high-dimensional, continuous latent space of the recurrence is beneficial in terms of capabilities, we note that there is concern that this comes with costs in model oversight in comparison to verbalized chains of thought, that are currently still human-readable. We provide initial results in Section 6 showing that the high-dimensional state trajectories of our models can be analyzed and some of their mechanisms interpreted. A number of additional visualization of the latent state can be found in the pages of the appendix. Ultimately, from our perspective, this type of thinking in latent space is not dissimilar from the latent computations that already happen in the intermediate layers of current, fixed-depth transformers. The increasing complexity of capability of these models necessitates white-box interpretability even without considering latent thinking, especially as verbal chain-of-thought traces from frontier models are becoming less interpretable. Analysis techniques that work to understand computations in intermediate layers also apply analogously to latent thinking.

## B   Future Work

Aside from work extending and analyzing the scaling behaviors of recurrent depth models, there are many questions that remain unanswered. For example, to us, there are potentially a large number of

novel post-training schemes that further enhance the capabilities of these models, such as fine-tuning to compress the recurrence or reinforcement learning with data with different hardness levels (Zelikman et al., 2024), or to internalize reasoning from CoT data into the recurrence (Deng et al., 2024).

Another aspect not covered in this work is the relationship to other modern architecture improvements. Efficient sequence mixing operations, especially those that are linear in sequence dimension, such as linear attention (Katharopoulos et al., 2020; Yang et al., 2024b), are limited in the number of comparisons that can be made. However, with recurrent depth, blocks containing linear operators can repeat until all necessary comparisons between sequence elements are computed (Suzgun et al., 2019), and recent work in this direction can be found in Schöne et al. (2025). For simplicity, we also focus on a single recurrence, where prior work has considered multiple successive recurrent stages (Takase and Kiyono, 2023; Csordás et al., 2024).

Finally, the proposed architecture is set up to be *compute-heavy*, with more "materialized" parameters than there are actual parameters. This naturally mirrors mixture-of-expert models (MoE), which are *parameter-heavy*, using fewer active parameters per forward pass than exist within the model (Shazeer et al., 2017; Fedus et al., 2022). We posit that where the recurrent-depth setup excels at learning reasoning patterns, the MoE excels at effectively storing and retrieving complex information. Their complementarity supports the hypothesis that a future architecture would contain both modifications. While in a standard MoE model, each expert can only be activated once per forward pass, or skipped entirely, a recurrent MoE model could also refine its latent state over multiple iterations, potentially routing to the same expert multiple times, before switching to a different one (Tan et al., 2023; Csordás et al., 2024). While MoE models are the currently leading solution to implement this type of "memory" in dense transformers, these considerations also hold for other memory mechanisms suggested for LLMs (Sukhbaatar et al., 2019; Fan et al., 2021; Wu et al., 2022; He et al., 2024).

## C    Training a large-scale recurrent-depth Language Model

In this section we provide a comprehensive report of the large-scale training run that we executed for this work. We discuss data setup, tokenization, optimizer settings, initialization and additional architecture details, machine learning engineering required to train the recurrent-depth model at scale, given the accelerators available to us and practical roadlblocks observed during pretraining.

### C.1    Training Setup

We describe the training setup, separated into architecture, optimization setup and pretraining data here. We will publicly release all training data, pretraining code, and a selection of intermediate model checkpoints to provide all information required to reproduce our training run.

**Pretraining Data.**    Given access to only enough compute for a single large scale model run, we opted for a dataset mixture that maximized the potential for emergent reasoning behaviors, not necessarily for optimal benchmark performance. Our final mixture is heavily skewed towards code and mathematical reasoning data with (hopefully) just enough general webtext to allow the model to acquire standard language modeling abilities. All sources are publicly available. We provide an overview in Figure 7. Following Allen-Zhu and Li (2024), we directly mix relevant instruction data into the pretraining data. However, due to compute and time constraints, we were not able to ablate this mixture. We expect that a more careful data preparation could further improve the model's performance. We list all data sources in Appendix F.

**Tokenization and Packing Details.**    We construct a vocabulary of 65536 tokens via BPE (Sennrich et al., 2016), using the implementation of Dagan (2024). In comparison to conventional tokenizer training, we construct our tokenizer directly on the instruction data split of our pretraining corpus, to maximize tokenization efficiency on the target domain. We also substantially modify the pre-tokenization regex (e.g. of Dagan et al. (2024)) to better support code, contractions and LaTeX. We include a `<|begin_text|>` token at the start of every document. After tokenizing our pretraining corpus, we pack our tokenized documents into sequences of length 4096. When packing, we discard document ends that would otherwise lack previous context, to fix an issue described as the "grounding problem" in Ding et al. (2024), aside from several long-document sources of mathematical content, which we preserve in their entirety.

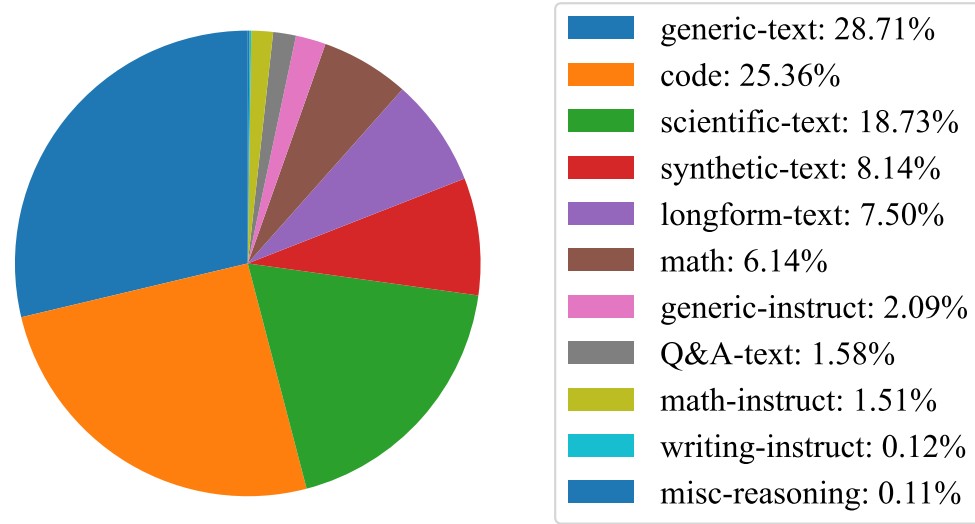

**Figure 7:** Distribution of data sources that are included during training. The majority of our data is comprised of generic web-text, scientific writing and code.

**Architecture and Initialization.** We scale the architecture described in Section 3, setting the layers to $(2, 4, 2)$, and train with a mean recurrence value of $\bar{r} = 32$. We mainly scale by increasing the hidden size to $h = 5280$, which yields $55$ heads of size of $96$. The MLP inner dimension is $17920$ and the RMSNorm $\varepsilon$ is $10^{-6}$. Overall this model shape has about $1.5$B parameters in non-recurrent prelude and head, $1.5$B parameters in the core recurrent block, and $0.5$B in the tied input embedding.

At small scales, most sensible initialization schemes work. However, at larger scales, we use the initialization of Takase et al. (2024) which prescribes a variance of $\sigma_h^2 = \frac{2}{5h}$. We initialize all parameters from a truncated normal distribution (truncated at $3\sigma$) with this variance, except all out-projection layers, where the variance is set to $\sigma_{\text{out}}^2 = \frac{1}{5hl}$, for $l = l_P + \bar{r}l_R + l_C$ the number of effective layers, which is $132$ for this model. As a result, the out-projection layers are initialized with fairly small values (Goyal et al., 2018). The output of the embedding layer is scaled by $\sqrt{h}$. To match this initialization, the state $s_0$ is also sampled from a truncated normal distribution, here with variance $\sigma_s^2 = \frac{2}{5}$.

**Locked-Step Sampling.** To enable synchronization between parallel workers, we sample a single depth $r$ for each micro-batch of training, which we synchronize across workers (otherwise workers would idle while waiting for the model with the largest $r$ to complete its backward pass). We verify at small scale that this modification improves compute utilization without impacting convergence speed, but note that at large batch sizes, training could be further improved by optimally sampling and scheduling independent steps $r$ on each worker, to more faithfully model the expectation over steps in Equation (1).

**Optimizer and Learning Rate Schedule.** We train using the Adam optimizer with decoupled weight regularization ($\beta_1 = 0.9$, $\beta_2 = 0.95$, $\eta = 5 \times 10^{-4}$) (Kingma and Ba, 2015; Loshchilov and Hutter, 2017), modified to include update clipping (Wortsman et al., 2023b) and removal of the $\varepsilon$ constant as in Everett et al. (2024). We clip gradients above 1. We train with warm-up and a constant learning rate (Zhai et al., 2022; Geiping and Goldstein, 2023), warming up to our maximal learning rate within the first $4096$ steps.

## C.2 Compute Setup and Hardware

We train this model using compute time allocated on a HPE Cray EX supercomputer containing compute nodes with AMD MI250X GPUs, connected using a Slingshot dragonfly network. The

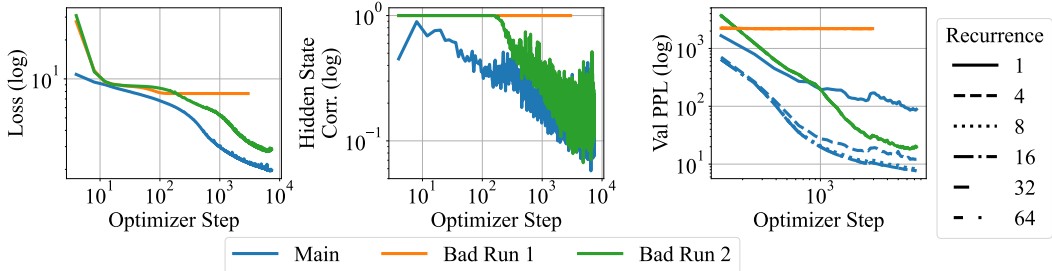

**Figure 8:** Plots of the initial 10000 steps for the first two failed attempts and the final, successful run ("Main"). Note the hidden state collapse (middle) and collapse of the recurrence (right) in the first two failed runs, underlining the importance of our architecture and initialization in inducing a recurrent model and explain the underperformance of these runs in terms of pretraining loss (left).

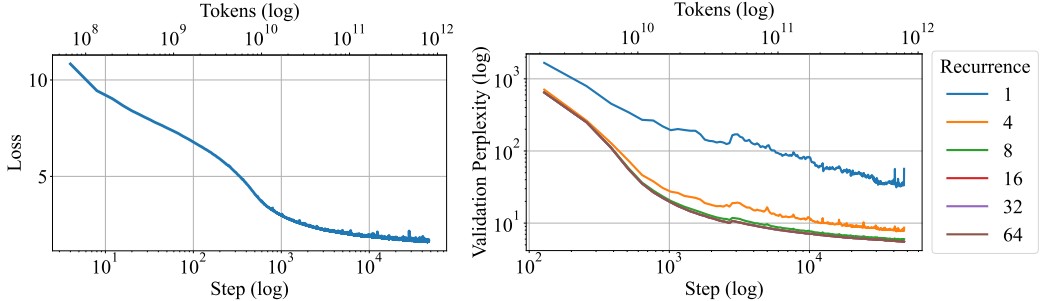

**Figure 9:** Left: Plot of pretrain loss over the 800B tokens on the main run. Right: Plot of val ppl at recurrent depths 1, 4, 8, 16, 32, 64. During training, the model improves in perplexity on all levels of recurrence.

scheduling system is orchestrated through SLURM. We train in bfloat16 mixed precision using a PyTorch-based implementation (Zamirai et al., 2021).

**Device Speed and Parallelization Strategy.** Nominally, each MI250X chip[2] achieves 192 TFLOP per GPU (AMD, 2021). For a single matrix multiplication, we measure a maximum achievable speed on these GPUs of 125 TFLOP/s on our software stack (ROCM 6.2.0, PyTorch 2.6 pre-release 11/02) (Bekman, 2023). Our implementation, using extensive PyTorch compilation and optimization of the hidden dimension to $h = 5280$ achieves a single-node training speed of 108.75 TFLOP/s, i.e. 87% AFU ("Achievable Flop Utilization"). Due to the weight sharing inherent in our recurrent design, even our largest model is still small enough to be trained using only data (not tensor) parallelism, with only optimizer sharding (Rajbhandari et al., 2020) and gradient checkpointing on a per-iteration granularity. With a batch size of 1 per GPU we end up with a global batch size of 16M tokens per step, minimizing inter-GPU communication bandwidth.

When we run at scale on 4096 GPUs, we achieve 52-64 TFLOP/s per GPU, i.e. 41%-51% AFU, or 1-1.2M tokens per second. To achieve this, we wrote a hand-crafted distributed data parallel implementation to circumvent a critical AMD interconnect issue, which we describe in more detail in Appendix C.4. Overall, we believe this may be the largest language model training run to completion in terms of number of devices used in parallel on an AMD cluster, as of time of writing.

**Training Timeline.** Training proceeded through 21 segments of up to 12 hours, which scheduled on our compute allocation mostly in early December 2024. We also ran a baseline comparison, where we train the same architecture but in a feedforward manner with only 1 pass through the core/recurrent block. This trained with the same setup for 180B tokens on 256 nodes with a batch size of 2 per GPU. Ultimately, we were able to schedule 795B tokens of pretraining of the main model. Due to our constant learning rate schedule, we were able to add additional segments "on-demand", when an allocation happened to be available.

---

[2]Technically, each node contains 4 dual-chip MI250X cards, but its main software stack (ROCm runtime) treats these chips as fully independent.

## C.3 Importance of Norms and Initializations at Scale

At small scales all normalization strategies worked, and we observed only tiny differences between initializations. The same was not true at scale. The first training run we started was set up with the same block sandwich structure as described above, but parameter-free RMSNorm layers, no embedding scale $\gamma$, a parameter-free adapter $A(\mathbf{s}, \mathbf{e}) = \mathbf{s} + \mathbf{e}$, and a peak learning rate of $4 \times 10^{-4}$. As shown in Figure 8, this run ("Bad Run 1", orange), quickly stalled.

While the run obviously stopped improving in training loss (left plot), we find that this stall is due to the model's representation collapsing (Noci et al., 2022). The correlation of hidden states in the token dimension quickly goes to 1.0 (middle plot), meaning the model predicts the same hidden state for every token in the sequence. We find that this is an initialization issue that arises due to the recurrence operation. Every iteration of the recurrence block increases token correlation, mixing the sequence until collapse.

We attempt to fix this by introducing the embedding scale factor, switching back to a conventional pre-normalization block, and switching to the learned adapter. Initially, these changes appear to remedy the issue. Even though token correlation shoots close to 1.0 at the start ("Bad Run 2", green), the model recovers after the first 150 steps. However, we quickly find that this training run is not able to leverage test-time compute effectively (right plot), as validation perplexity is the same whether 1 or 32 recurrences are used. This initialization and norm setup has led to a local minimum as the model has learned early to ignore the incoming state $\mathbf{s}$, preventing further improvements.

In a third, and final run ("Main", blue), we fix this issue by reverting back to the sandwich block format, and further dropping the peak learning rate to $4 \times 10^{-5}$. This run starts smoothly, never reaches a token correlation close to 1.0, and quickly overtakes the previous run by utilizing the recurrence and improving with more iterations.

With our successful configuration, training continues smoothly for the next 750B tokens without notable interruptions or loss spikes. We plot training loss and perplexity at different recurrence steps in Figure 9. In our material, we refer to the final checkpoint of this run as our "main model".

## C.4 Additional Implementation Details

**Device Speed Details**    Nominally, each MI250X (AMD, 2021) achieves 383 TFLOP in bfloat16, i.e. 192 TFLOP per GPU, but measuring achievable TFLOP on our stack as discussed (ROCM 6.2.0, PyTorch 2.6 pre-release 11/02) for arbitrary matrix multiplication shapes (i.e. we measure the peak achievable speed of the best possible shape iterating over shapes between 256 and 24576 in intervals of 256 and 110 (Bekman, 2023)), we measure a peak of 125 TFLOP/s on the nodes we are provided. Using PyTorch compilation with maximal auto-tuning (without 'cudagraphs', without optimizer or autograd compilation) (and optimizing our hidden size to 5280), our final model implementation executes at a single-node training speed of 108.75 TFLOP/s, i.e. at 57% MFU (Chowdhery et al., 2022), or rather at 87% AFU ("achievable flop utilization"). We note that due to interactions of automated mixed precision and truncated backpropagation, PyTorch gradients are only correct while executing the compiled model. We further circumvent issues with the flash attention implementation shipped with PyTorch sdpa using the AMD fork of the original flash attention repository[3], which can be found at https://github.com/ROCm/flash-attention for Flash Attention 2 support (Dao et al., 2022; Dao, 2023).

**Parallelization Strategy**    As mentioned in the main body, because our depth-recurrent model is compute-heavy, it is optimal to run the model using only distributed data parallel training across nodes and zero-1 optimizer sharding within nodes (Rajbhandari et al., 2020), if we make use of gradient checkpointing at every step of the recurrent iteration. This allows us to eschew more communication-heavy parallelization strategies that would be required for models with the same FLOP footprint, but more parameters, which require substantial planning on this system (Singh et al., 2024; Singh and Bhatele, 2022). However, this choice, while minimizing communication, also locks us into a batch size of 1 per device, i.e. 4096 in total, and 16M tokens per step.

---

[3]https://github.com/Dao-AILab/flash-attention/

**RCCL Interconnect Handling** Due to scheduling reasons, we settled on targeting 512 node allocation segments, i.e. 4096 GPUs. However, this posed a substantial network interconnect issue. The connection speed between nodes is only acceptable, if RCCL (AMD GPU communication collectives) commands are routed through open fabrics interface calls, which happens via a particular plugin[4]. To achieve sufficient bus bandwidth above 100GB/s requires `NCCL_NET_GDR_LEVEL=PHB`, a setting that, on NVIDIA systems, allows packages to go through the CPU, and only uses direct interconnect if GPU and NIC are on the same (NUMA) node (Wu and Stock, 2024). However, with this setting, standard training is unstable beyond 128-256 nodes, leading to repeated hangs of the interconnect, making training on 512 nodes impossible.

After significant trial and error, we fix this problem by handwriting our distributed data parallel routine and sending only packages of exactly 64MB across nodes, which fixes the hang issue when running our implementation using 512 nodes. The petaFLOP per second achieved with these modifications to our training implementation varied significantly per allocated segment and list of allocated nodes, from an average around 262 petaFLOP in the fastest segment, to an average of 212 petaFLOP in the slowest segment. This is a range of 52-64 TFLOP/s per GPU, i.e. 41%-51% AFU, or 1-1.2M tokens per second.

**Pretraining Metrics.** During the pretraining run, we run a careful tracking of optimizer and model health metrics, tracking effective Adam learning rates per layer, optimizer RMS (Wortsman et al., 2023a), $L^2$ and $L^1$ parameter and gradient norms, recurrence statistics such as $\frac{||s_k - s_{k-1}||}{||s_k||}$, $||s_k||$, $||s_0 - s_k||$. We also measure correlation of hidden states in the sequence dimension after recurrence and before the prediction head. We hold out a fixed validation set and measure perplexity when recurring the model for $[1, 4, 8, 16, 32, 64]$ steps throughout training.

**Ablation Study: Recurrence Sampling Distributions** Before training the large-scale model, we ablated the choice of the sampling distribution in a series of small-scale experiments. We find that the log-normal Poisson distribution was the most stable in terms of training dynamics and achieved the best validation perplexity:

| Sampling Scheme | Validation PPL |
| --- | --- |
| Log-normal Poisson | 12.97 |
| Irwin-Hall | 12.99 |
| Schwarzschild-Bansal | 13.16 |
| Exponential | 13.26 |
| Gamma | 13.31 |
| Geometric | 13.33 |
| Uniform | 13.33 |

**Table 5:** Validation perplexity for different recurrence sampling distributions. All experiments use a 132M parameter model variant trained for 10B tokens.

Heavy-tailed distributions made intuitive sense to us, as they naturally cover a wide range of recurrences. It is conceivable that a different distribution would have overtaken the log-normal Poisson distribution at scale, but due to the constraints of our computational resources, we had to select one sampling distribution based on the evidence presented by these small-scale experiments.

# D  Additional Evaluations and Benchmark Instructions

## D.1  Benchmarking Settings

For all evaluations, we run the model with temperature 0. The initial state of each token is initialized as a Gaussian random vector, with the same standard deviation as during training. When we describe a setting by, e.g. $r = 32$, that means that to control recurrence, we run all queries with exactly 32 recurrences. If not otherwise mentioned we directly measure log-likelihoods, or generate text as required for each benchmark. We run all evaluations in pure `bfloat16`.

---

[4] https://github.com/ROCm/aws-ofi-rccl

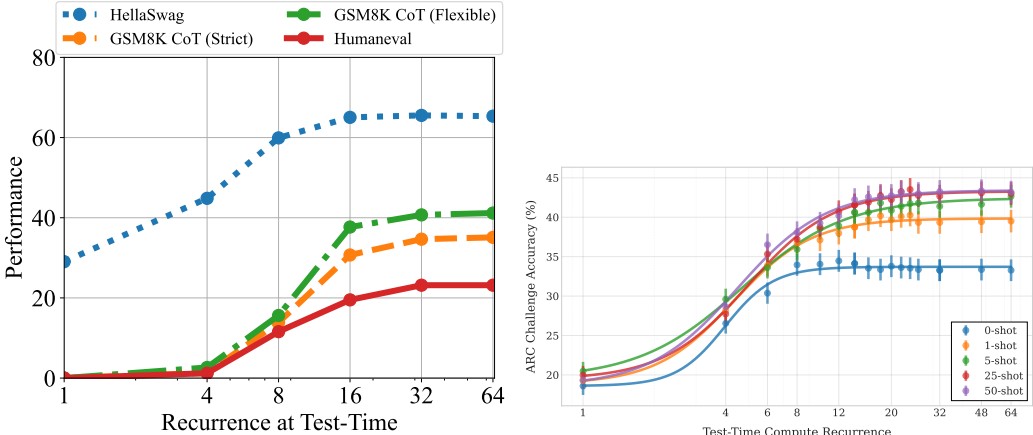

**Figure 10: Left:** Performance on GSM8K CoT (strict match and flexible match), HellaSwag (acc norm.), and HumanEval (pass@1). As we increase compute, the performance on these benchmarks increases. HellaSwag only needs 8 recurrences to achieve near peak performance while other benchmarks make use of more compute. **Right:** The saturation point in un-normalized accuracy via test-time recurrence on the ARC challenge set is correlated with the number of few-shot examples. The model uses more recurrence to extract more information from the additional few-shot examples, making use of more compute if more context is given.

In a few tables we also report accuracy with chat formats. In those cases we always run with the same chat template as seen during training, which is

```
{% set loop_messages = messages %}
{% for message in loop_messages %}
{% set start_content = '<|begin_header|>' %}
{% set end_content = message['content'] | trim + '<|end_turn|>' %}
{% if loop.index0 == 0 %}
{% set start_content = bos_token + start_content %}
{% endif %}
{% if message['role'] == 'Huginn' or message['role'] == 'assistant' %}
{% set start_content = start_content + 'Huginn<|end_header|>\n\n' %}
{{ start_content }}{% generation %}{{ end_content }}{% endgeneration %}
{% else %}
{% set start_content = start_content + message['role'] + '<|end_header|>\n\n' %}
{{ start_content }}{{ end_content }}{% endif %}{% endfor %}
{% if add_generation_prompt %}
{{ '<|begin_header|>Huginn<|end_header|>\n\n' }}
{% else %}{{ '<|end_text|>' }}{
% endif %}
```

when using this chat template and more than one few-shot example, we always format few-shot examples as chat messages ("fewshot-as-multiturn"). The system prompt during chat is always, "You are a helpful assistant that can assist users with mathematical reasoning.".

Using our model implementation (see code release), all evaluations can be replicated in the lm-eval harness via, e.g.

```
lm_eval --model hf --model_args pretrained=tomg-group-umd/huginn-0125,mean_recurrence=32,trust_remote_code=True,
dtype=bfloat16,max_length=4096  --tasks mmlu --batch_size=auto --num_fewshot=5
--output_path=outputs/misc --gen_kwargs=max_length=4096,max_gen_toks=1024
```

for MMLU (5-shot) with $r = 32$, and similarly for all other benchmarks evaluating log-likelihoods, or

```
lm_eval --model hf --model_args pretrained=tomg-group-umd/huginn-0125,trust_remote_code=True,
dtype=bfloat16,mean_recurrence=32  --tasks gsm8k_cot --batch_size=auto --num_fewshot=8
--output_path=outputs/gsm8k --apply_chat_template=True
--system_instruction="You are a helpful assistant that can assist users with mathematical reasoning."
--fewshot_as_multiturn
```

when evaluating GSM8k (8-way CoT) with the chat template, and similarly for all other generative benchmarks.

**Table 6:** Comparison of Open and Closed QA Performance (%) (Mihaylov et al., 2018). In the open exam, a relevant fact is provided before the question is asked. In this setting, our smaller model closes the gap to other open-source models, indicating that the model is capable, but has fewer facts memorized.

| Model | Closed | Open | $\Delta$ |
|---|---|---|---|
| Amber | 41.0 | 46.0 | +5.0 |
| Pythia-2.8b | 35.4 | 44.8 | +9.4 |
| Pythia-6.9b | 37.2 | 44.2 | +7.0 |
| Pythia-12b | 35.4 | 48.0 | +12.6 |
| OLMo-1B | 36.4 | 43.6 | +7.2 |
| OLMo-7B | 42.2 | 49.8 | +7.6 |
| OLMo-7B-0424 | 41.6 | 50.6 | +9.0 |
| OLMo-7B-0724 | 41.6 | 53.2 | +11.6 |
| OLMo-2-1124 | 46.2 | 53.4 | +7.2 |
| Ours ($r = 32$) | 38.2 | 49.2 | +11.0 |

## D.2 Recurrence Scaling by Tasks

Further, we chart the improvement as a function of test-time compute on several of these tasks for the main model in Figure 10. We find that saturation is highly task-dependent, on easier tasks the model saturates quicker, whereas it benefits from more compute on others.

## D.3 Recurrence and Context

We evaluate ARC-C performance as a function of recurrence and number of few-shot examples in the context in Figure 10. Interestingly, without few-shot examples to consider, the model saturates in compute around 8-12 iterations. However, when more context is given, the model can reason about more information in context, which it does, saturating around 20 iterations if 1 example is provided, and 32 iterations, if 25-50 examples are provided, mirroring generalization improvements shown for recurrence (Yang et al., 2024a; Fan et al., 2025). Similarly, we see that if we re-evaluate OBQA in Table 6, but do not run the benchmark in the default lm-eval "closed-book" format and rather provide a relevant fact, our recurrent model improves significantly almost closing the gap to OLMo-2. Intuitively this makes sense, as the recurrent models has less capacity to memorize facts but more capacity to reason about its context.

## D.4 Comparison to Open-Weight Models

A comparison to open-weight models can be found in Table 7, note that these models are mostly trained on 10-15x more compute.

## D.5 Regarding KV-cache Misses in Token-Adaptive Early Exits

Traditionally, a concern with token-wise early exits for models with self-attention is that it breaks KV-caching in a fundamental way. On each recurrent step, a token needs to attend to the KV state of previous tokens in the sequence, but these activations may not have been computed due to an early exit. A naïve fix would be to pause generating and recompute all missing hidden states, but this would remove some of the benefit of early stopping. Instead, as in Elbayad et al. (2019), we attend to the last, deepest available KV states in the cache. Because all recurrent KV cache entries are generated by the same K,V projection matrices from successive hidden states, they "match", and therefore the model is able to attend to the latest cache entry from every previous token, even if computed at different recurrent depths.

## D.6 Zero-Shot Continuous Chain-of-Thought

By attending to the output of later steps of previous tokens in the early steps of current tokens, as described in the KV-cache sharing section, we actually construct a computation that is deeper than the current number of recurrence steps. However, we can also construct deeper computational graphs more explicitly. Instead of sampling a random initial state $s_0$ at every generation step, we can

**Table 7:** Results on lm-eval-harness tasks across various open-weight models. We show ARC (Clark et al., 2018), HellaSwag (Zellers et al., 2019), MMLU (Hendrycks et al., 2021a), OpenBookQA (Mihaylov et al., 2018), PiQA (Bisk et al., 2020), SciQ (Johannes Welbl, 2017), WinoGrande (Sakaguchi et al., 2021), and GSM8k (zero-shot).

| Model | Param | Tokens | ARC-E | ARC-C | HellaSwag | MMLU | OBQA | PiQA | SciQ | WinoGrande | GSM8k | GSM8k CoT |
|---|---|---|---|---|---|---|---|---|---|---|---|---|
| random | | | 25.0 | 25.0 | 25.0 | 25.0 | 25.0 | 50.0 | 25.0 | 50.0 | 0.0/0.0 | 0.0/0.0 |
| Qwen2.5-0.5B | 0.5B | 18T | 64.60 | 29.27 | 40.53 | 47.29 | 24.60 | 70.29 | 93.00 | 56.35 | 0.08/4.78 | 37.76/40.94 |
| Qwen2.5-1.5B | 1.5B | 18T | 75.38 | 41.47 | 50.19 | 59.75 | 32.00 | 75.79 | 94.10 | 63.22 | 0.00/9.48 | 55.88/67.10 |
| Qwen2.5-3B | 3B | 18T | 77.23 | 44.54 | 55.02 | 65.05 | 29.40 | 78.18 | 96.10 | 67.80 | 0.23/5.84 | 59.36/75.51 |
| Llama-3.2-3B | 3B | 9T | 74.58 | 42.58 | 55.20 | 54.13 | 31.00 | 76.77 | 95.60 | 69.22 | 0.00/11.30 | 29.04/30.71 |
| Llama-3.1-8B | 8B | 15T | 81.73 | 51.02 | 59.99 | 63.32 | 33.00 | 80.20 | 96.40 | 73.72 | 0.00/25.70 | 54.51/56.79 |
| Ministral-8B | 8B | - | 81.57 | 54.35 | 59.67 | 63.99 | 36.00 | 81.01 | 96.80 | 73.56 | 0.00/65.96 | 73.09/76.42 |
| Qwen3-4B-Instruct | 4B | 36T | 83.08 | 58.62 | 69.14 | 70.55 | 40.20 | 76.06 | 95.90 | 67.80 | 0.00/71.65 | 47.54/55.65 |
| Ours, ($r = 32$) | 3.5B | 0.8T | 69.91 | 38.23 | 65.21 | 31.38 | 38.80 | 76.22 | 93.50 | 59.43 | 24.87/38.13 | 34.80/42.08 |
| Merged, ($r = 32$) | 3.5B | 0.8T | 68.22 | 32.85 | 49.39 | 29.26 | 26.20 | 75.35 | 93.50 | 59.35 | 0.02/30.17 | 37.45/ 46.85 |
| Merged, ($r = 64$) | 3.5B | 0.8T | 68.01 | 32.94 | 49.50 | 29.19 | 26.00 | 75.30 | 93.60 | 58.33 | 0.02/31.46 | 38.59/47.23 |

**Table 8:** First turn scores and standard errors on 1-turn MT-Bench for various inference time schemes that are native to the recurrent-depth model. Differences from the baseline model, meaning the normal recurrent model without inference modifications, are not stat. significant.

| Model | MT-Bench | Std. Error |
|---|---|---|
| cache compression, $s = 4$ | 5.856 | 0.395 |
| baseline, 64 iterations | 5.693 | 0.386 |
| cache compression, $s = 16$ | 5.687 | 0.402 |
| baseline, 32 iterations | 5.662 | 0.388 |
| cache compression, $s = 8$ | 5.631 | 0.384 |
| KL exit, $t = 5 \times 10^{-4}$ | 5.562 | 0.389 |

warm-start with the last state $s_r$ from the previous token. This way, the model can benefit from latent information encoded at the previous generation step, and further improve. As shown in Figure 4, this reduces the average number of steps required to converge by 1-2. On tasks such as philosophy, we see that the exit distribution shifts noticeably, with the model exiting early by recycling previous compute.

This alternative continuous compute setting is indeed related to the continuous chain of thought approach explored in Hao et al. (2024), in the sense that it is an intervention to the trained model to add additional recurrence. To achieve a similar behavior in fixed-depth transformers, Hao et al. (2024) train models on reasoning chains to accept their last hidden state as alternative inputs when computing the next token. Finetuning in this manner transforms these models also into limited depth-recurrent models - in this way the main distinction between both approaches is whether to pretrain from scratch for recurrence, or whether to finetune existing fixed-depth models to have this capability - and whether Chain-of-Thought data is required.

### D.7 Saturation Evaluation

We provide additional analysis of the saturation points for a range of benchmarks in Figure 12 and Figure 13, showing that how quickly the model solves the task is highly data-dependent, with a substantial number of trivia and language questions from tasks such as SciQ, BLiMP (Warstadt et al., 2020) being solvable without recurrence. Further, many easy benchmark are solvable with 4 or less recurrences. On the other hand, reasoning tasks, such as HumanEval, GSM8k and Mastermind-Eval (Golde et al., 2025), but also deduction-heavy benchmarks such as MMLU or BBH (Suzgun et al., 2022) continue improving with additional compute, and sit at chance accuracy at 1 or 4 recurrences. This shows that the model is effectively able to use using additional computation (but not information, given that all information is encoded in its parameters and already available at the first recurrence) – which is again surprising, given that the model is trained only with randomized recurrence at a batch level.

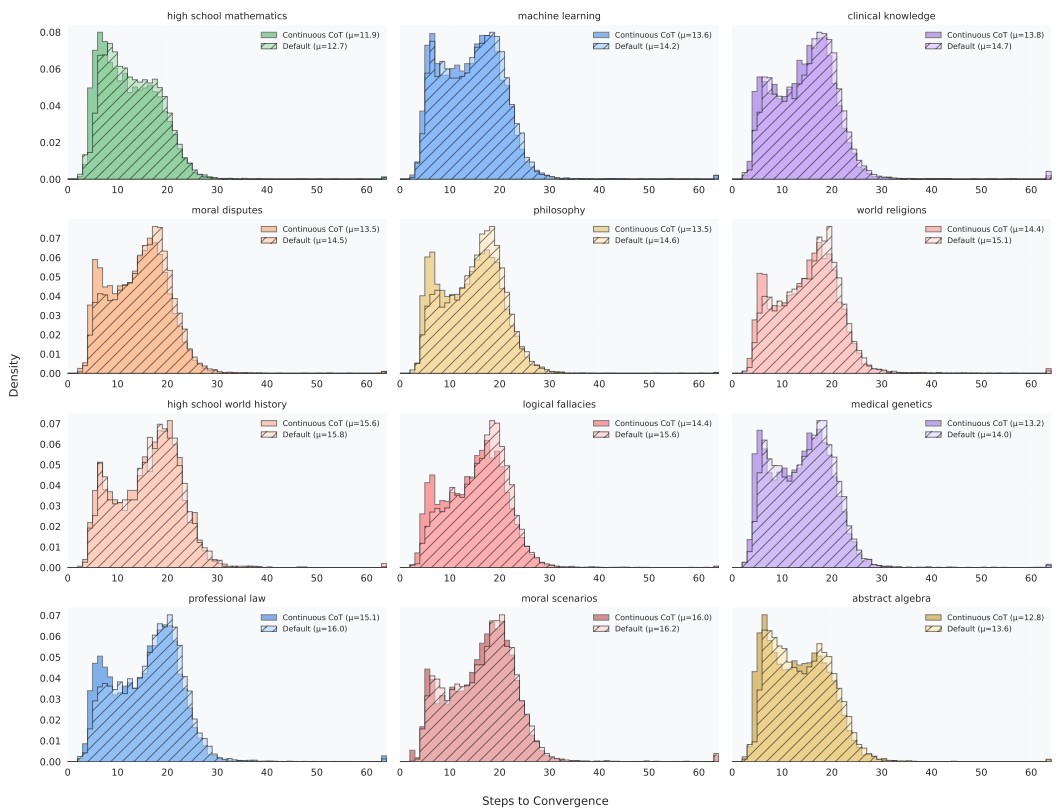

**Figure 11:** Additional categories for Figure 4 in the main body.

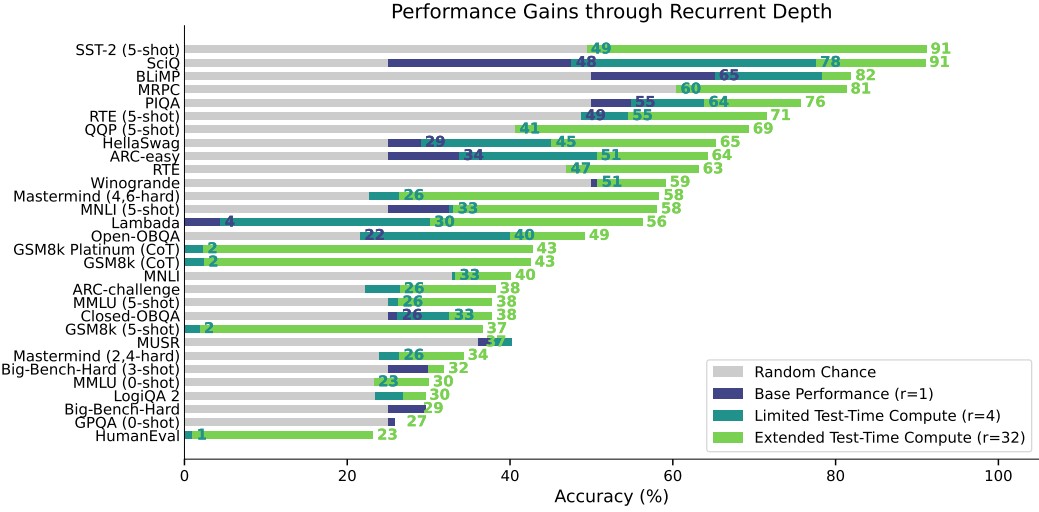

**Figure 12:** Accuracy at $r = 1$, $r = 4$, and $r = 32$ for a range of benchmarks, showing that accuracy saturation is highly context-dependent. Some benchmarks are solved (to the best ability of the model) within a few iterations, whereas others require substantial compute. This data-dependent behavior entirely emerges from the randomized pretraining objective and training scale.

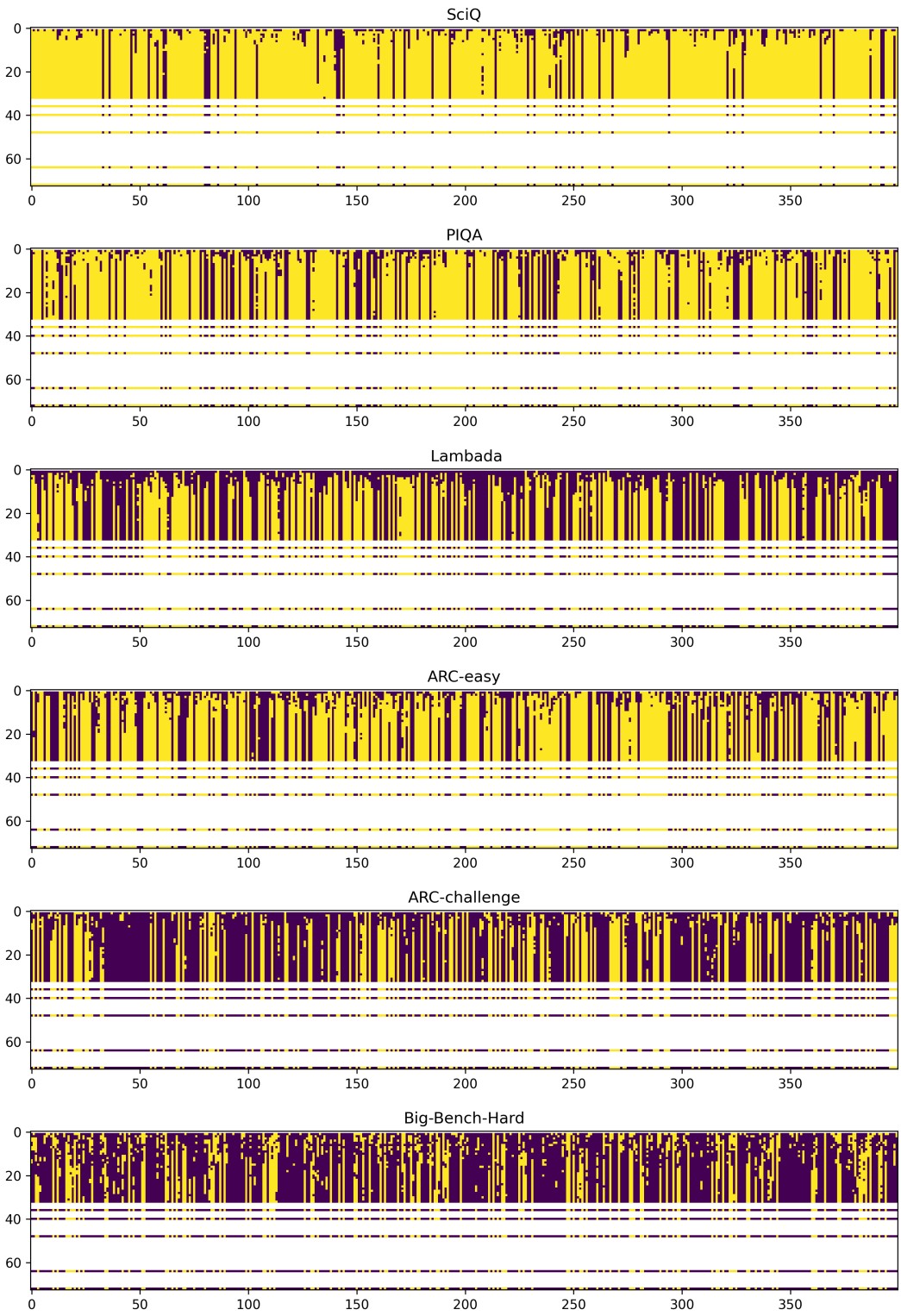

**Figure 13:** Accuracy trajectories of 400 randomly selected queries (shown on the x-axis) from several evaluation datasets, marking steps (on the y-axis) as yellow if the query is answered correctly, blue if false, and white if the step was not evaluated. Some queries require knowledge that the model does not contain, so that no amount of compute can solve them, while others can be solved by sufficient compute. Exploration behavior is strongly data-dependent, comparing e.g. Big-Bench, where early accuracy patterns are almost chaotic, and SciQ, where accuracy in the same regime is noticeably smoother.

**Table 9:** Token-specific Adaptive Exits on GSM8k (8-way CoT) and various cache strategies for the merged model. Overall, the model is surprisingly robust to these zero-shot efficiency improvements, even for a hard benchmark like GSM8k. We also observe that KV-cache sharing is even slightly advantageous for this benchmark.

| Criterion | Threshold | KV Cache Strategy | Cont. Compute | Accuracy |
|---|---|---|---|---|
| *Baseline Configuration* | | | | |
| – | – | – | False | 46.63% |
| – | – | – | True | 46.63% |
| *KV-cache sharing* | | | | |
| – | – | latest-m4-compress-s16 | False | 47.16% |
| – | – | latest-m4-compress-s8 | True | 46.40% |
| – | – | latest-m4-compress-s4 | False | 47.08% |
| – | – | latest-m4-compress-s4 | True | 46.78% |
| *Latent-Diff Criterion* | | | | |
| latent-diff | 0.03 | latest-m4-compress-s16 | False | 46.78% |
| latent-diff | 0.03 | latest-m4 | False | 46.10% |
| latent-diff | 0.03 | latest-m4 | True | 46.10% |
| *Entropy-Diff Criterion* | | | | |
| entropy-diff | 0.0001 | latest-m4 | False | 46.17% |
| entropy-diff | 0.001 | latest-m4 | True | 44.28% |
| entropy-diff | 0.001 | latest-m4-compress-s16 | False | 44.05% |
| entropy-diff | 0.001 | latest-m4 | False | 41.55% |
| *KL Criterion* | | | | |
| kl | $1 \times 10^{-5}$ | latest-m4 | False | 45.87% |
| kl | 0.0005 | latest-m4-compress-s16 | False | 44.81% |
| kl | 0.0005 | latest-m4 | True | 44.58% |
| kl | 0.0005 | latest-m4 | False | 44.43% |
| *MinP-KL Criterion* | | | | |
| minp-kl | $5 \times 10^{-7}$ | latest-m4 | False | 43.52% |
| minp-kl | $1 \times 10^{-6}$ | latest-m4-compress-s16 | False | 42.30% |
| minp-kl | $1 \times 10^{-6}$ | latest-m4 | True | 41.02% |
| minp-kl | $1 \times 10^{-6}$ | latest-m4 | False | 40.71% |
| *Argmax-Stability Criterion* | | | | |
| argmax-stability | 10 | latest-m4 | False | 41.85% |
| argmax-stability | 5 | latest-m4 | True | 30.93% |
| argmax-stability | 5 | latest-m4 | False | 26.46% |
| argmax-stability | 5 | latest-m4 | False | 26.46% |

## D.8 Model Stability at Arbitrary Depths

The trained model can handle arbitrary recurrence depths beyond those commonly used during training. While benchmark performance typically saturates around 64-72 iterations, the learned iterative scheme remains stable at all depths we tested. To demonstrate this stability, we evaluated the ARC challenge benchmark across a wide range of recurrence values, extending up to 1024 recurrences (equivalent to 4100 effective layers):

| Recurrence | 32 | 64 | 128 | 256 | 512 | 1024 |
|---|---|---|---|---|---|---|
| **Effective Layers** | 132 | 260 | 516 | 1028 | 2052 | 4100 |
| **Norm. Acc. (%)** | 37.88 | 37.37 | 37.63 | 37.03 | 37.20 | 37.37 |

**Table 10:** Performance on ARC challenge benchmark across different recurrence depths. The accuracy fluctuations are within the range of typical noise for this benchmark, demonstrating the model's stability even at extreme recurrence counts.

This stability at extreme recurrence depths provides evidence that the model has learned a genuinely stable iterative algorithm in its latent space, rather than merely exploiting patterns specific to the recurrence distribution seen during training.

**Table 11:** Results on zero-shot evaluations across open-source models. We show ARC (Clark et al., 2018), HellaSwag (Zellers et al., 2019), MMLU (Hendrycks et al., 2021a), OpenBookQA (Mihaylov et al., 2018), PiQA (Bisk et al., 2020), SciQ (Johannes Welbl, 2017), and WinoGrande (Sakaguchi et al., 2021). We report normalized accuracy on PiQA, OBQA, ARC-C and HellaSwag. Inference output tok(ens)/sec measured via vllm benchmark, see Appendix D.9.

| Model | Param | Tokens | tok/sec | ARC-E | ARC-C | HellaSwag | MMLU | OBQA | PiQA | SciQ | WinoGr |
|---|---|---|---|---|---|---|---|---|---|---|---|
| random | | | | 25.0 | 25.0 | 25.0 | 25.0 | 25.0 | 50.0 | 25.0 | 50.0 |
| Amber | 7B | 1.2T | 963 | 65.70 | 37.20 | 72.54 | 26.77 | 41.00 | 78.73 | 88.50 | 63.22 |
| Pythia-2.8b | 2.8B | 0.3T | 502 | 58.00 | 32.51 | 59.17 | 25.05 | 35.40 | 73.29 | 83.60 | 57.85 |
| Pythia-6.9b | 6.9B | 0.3T | 1008 | 60.48 | 34.64 | 63.32 | 25.74 | 37.20 | 75.79 | 82.90 | 61.40 |
| Pythia-12b | 12B | 0.3T | 451 | 63.22 | 34.64 | 66.72 | 24.01 | 35.40 | 75.84 | 84.40 | 63.06 |
| OLMo-1B | 1B | 3T | 5575 | 57.28 | 30.72 | 63.00 | 24.33 | 36.40 | 75.24 | 78.70 | 59.19 |
| OLMo-7B | 7B | 2.5T | 917 | 68.81 | 40.27 | 75.52 | 28.39 | 42.20 | 80.03 | 88.50 | 67.09 |
| OLMo-7B-0424 | 7B | 2.05T | 917 | 75.13 | 45.05 | 77.24 | 47.46 | 41.60 | 80.09 | 96.00 | 68.19 |
| OLMo-7B-0724 | 7B | 2.75T | 917 | 74.28 | 43.43 | 77.76 | 50.18 | 41.60 | 80.69 | 95.70 | 67.17 |
| OLMo-2-1124 | 7B | 4T | 917 | 82.79 | 57.42 | 80.50 | 60.56 | 46.20 | 81.18 | 96.40 | 74.74 |
| OLMo-2-32B-0.8T | 32B | 0.8T | 283 | 79.46 | 54.69 | 79.85 | 59.49 | 48.00 | 81.07 | 92.70 | 75.69 |
| Ours, ($r = 4$) | 3.5B | 0.8T | 1327 | 49.07 | 27.99 | 43.46 | 23.39 | 28.20 | 64.96 | 80.00 | 55.24 |
| Ours, ($r = 8$) | 3.5B | 0.8T | 726 | 65.11 | 35.15 | 58.54 | 25.29 | 35.40 | 73.45 | 92.10 | 55.64 |
| Ours, ($r = 16$) | 3.5B | 0.8T | 380 | 69.49 | 37.71 | 64.67 | 31.25 | 37.60 | 75.79 | 93.90 | 57.77 |
| Ours, ($r = 32$) | 3.5B | 0.8T | 201 | 69.91 | 38.23 | 65.21 | 31.38 | 38.80 | 76.22 | 93.50 | 59.43 |

## D.9 Inference Speed Benchmark

To provide ballpark figures for how much compute time is required in practice to run the model at inference, we implement a minimal version of the architecture in `vllm` with KV-cache sharing, Appendix D.5, which we do not optimize for performance. With this comparison implementation in hand, we execute a controlled `vllm` throughput benchmark using random data with input length 512 and output length 512 using the V1 engine. We run the benchmark for 128 prompts and report output tokens/s measured on a single NVIDIA RTX 6000 Ada GPU, which we show in Table 11 (an extended version of Table 1). We note that for the 32B OLMo model we actually use cards for inference (due to model size), and so we report tokens/second per GPU for this model. We further note that the Pythia-2.8B model inference speed is unusually slow in our evaluation, as the model architecture does not trigger the flash attention backend, but the flex attention backend. All other models trigger the flash attention backend, although the head size of our recurrent-depth model is not necessarily an optimal shape for flash attention on NVIDIA GPUS, as the hidden dimension was optimized for the AMD-MI250X in particular.

## E  Latent Space Visualizations

On the next pages, we print a number of latent space visualizations in more details than was possible in Section 6. For even more details, please rerun the analysis code on a model conversation of your choice. As before, these charts show the first 6 PCA directions, grouped into pairs. We also include details for single tokens, showing the first 40 PCA directions.

**Path Independence.**   We verify that our models maintain path independence, in the sense of Anil et al. (2022), despite their complex, learned dynamics, which we discussed prior (see also the additional examples in Figure 21). When re-initializing from multiple starting states $s_0$, the model moves in similar trajectories, exhibiting consistent behavior. The same orbital patterns, fixed points, or directional drifts emerge regardless of initialization.

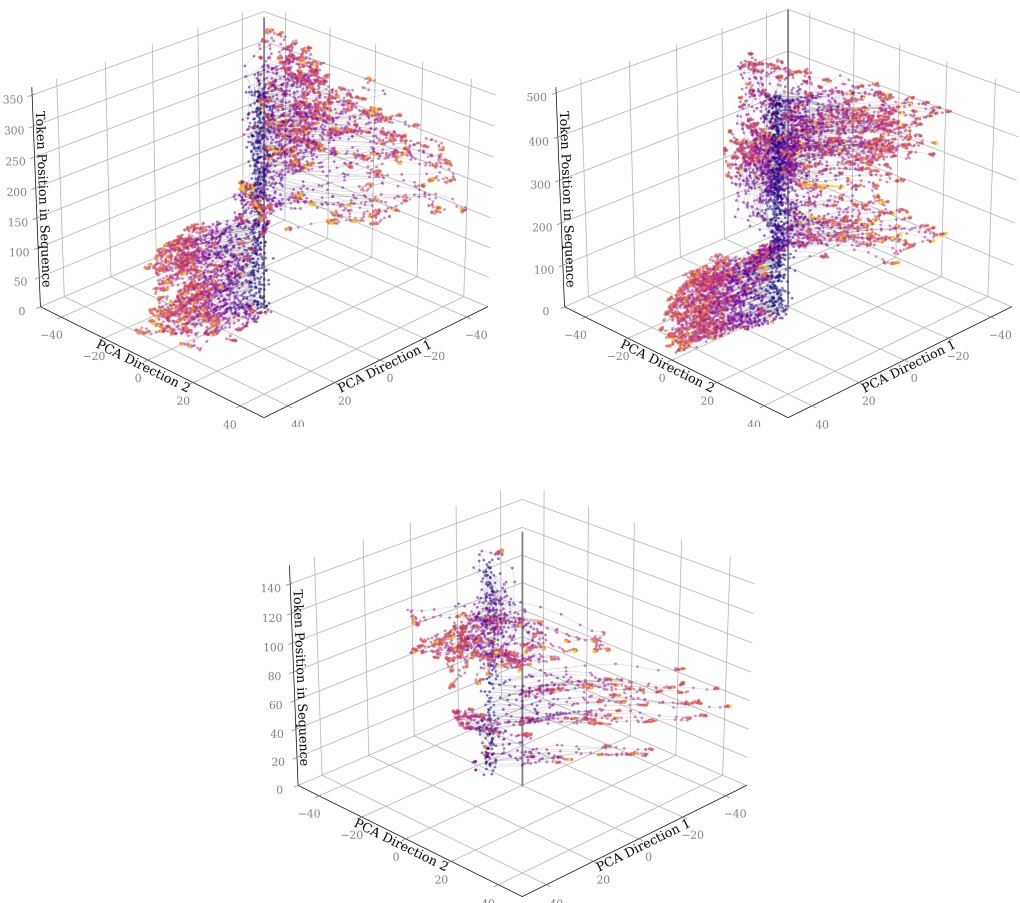

**Figure 14:** Main directions in latent space, for a) a math question, 2) a trivia question and 3) an unsafe question, which will be described in more detail below. *Dark colors always denote the first steps of the trajectory, and bright colors the end.* Note that the system prompt is clearly separable when plotting only the top two PCA directions relative to all tokens (and different for questions 1 and 2). Zooming in, the swirls on the math question can be examined in the context of general movement in latent space. More detailed visualizations follow on later pages.

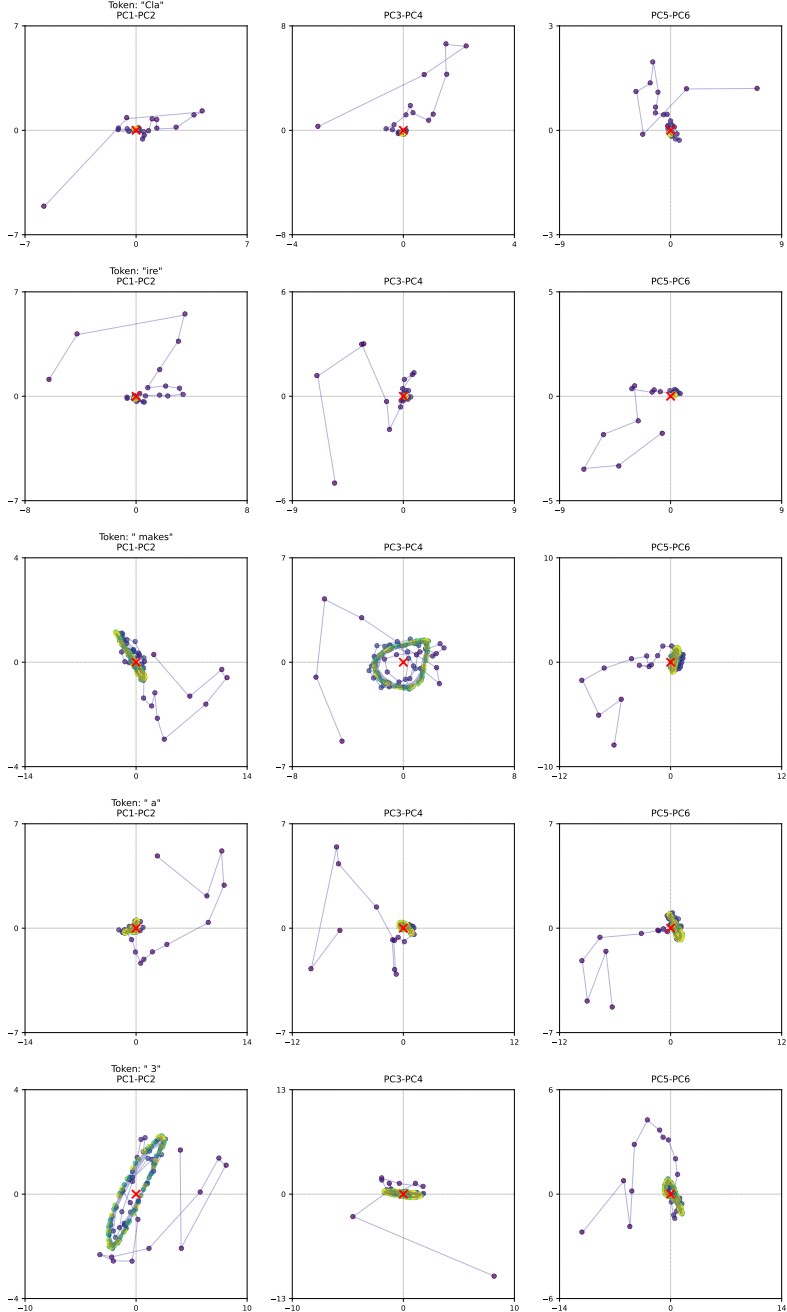

**Figure 15:** Latent Space trajectories for a math question. The model is rotating the number three, on which the problem hinges. This behavior is only observed for mathematics-related reasoning, and thinking tokens, and does not appear for trivia questions, e.g. as above. The question is `Claire makes a 3 egg omelet every morning for breakfast. How many dozens of eggs will she eat in 4 weeks?` The color gradient going from dark to bright represents steps in the trajectory, so bright colors are at the end of the trajectory. The center of mass is marked in red.

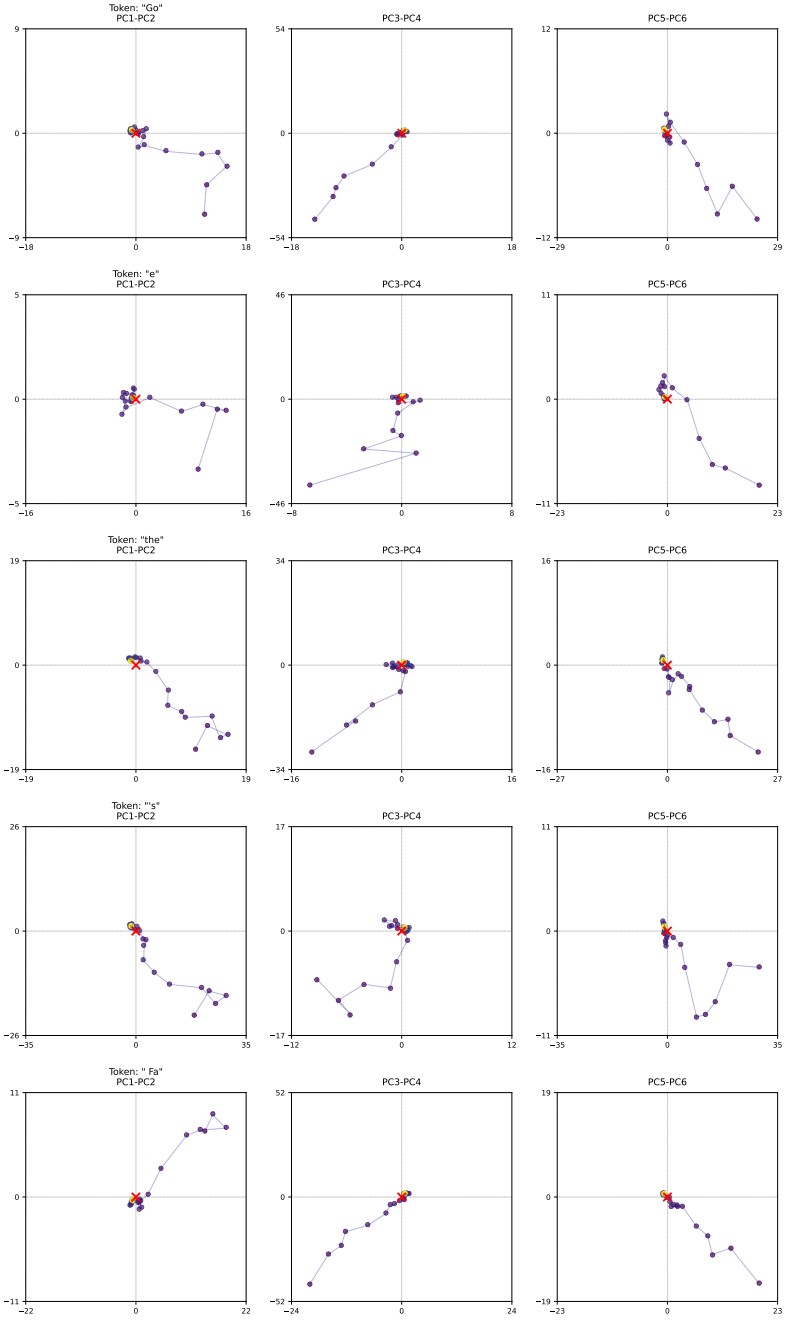

**Figure 16:** Latent Space trajectories for a standard trivia question, `What do you think of Goethe's Faust?`. Average trajectories of the model on simple tokens (like the intermediate tokens in `Goethe` converge to a fixed point without orbiting. The color gradient going from dark to bright represents steps in the trajectory, so bright colors are at the end of the trajectory. The center of mass is marked in red.

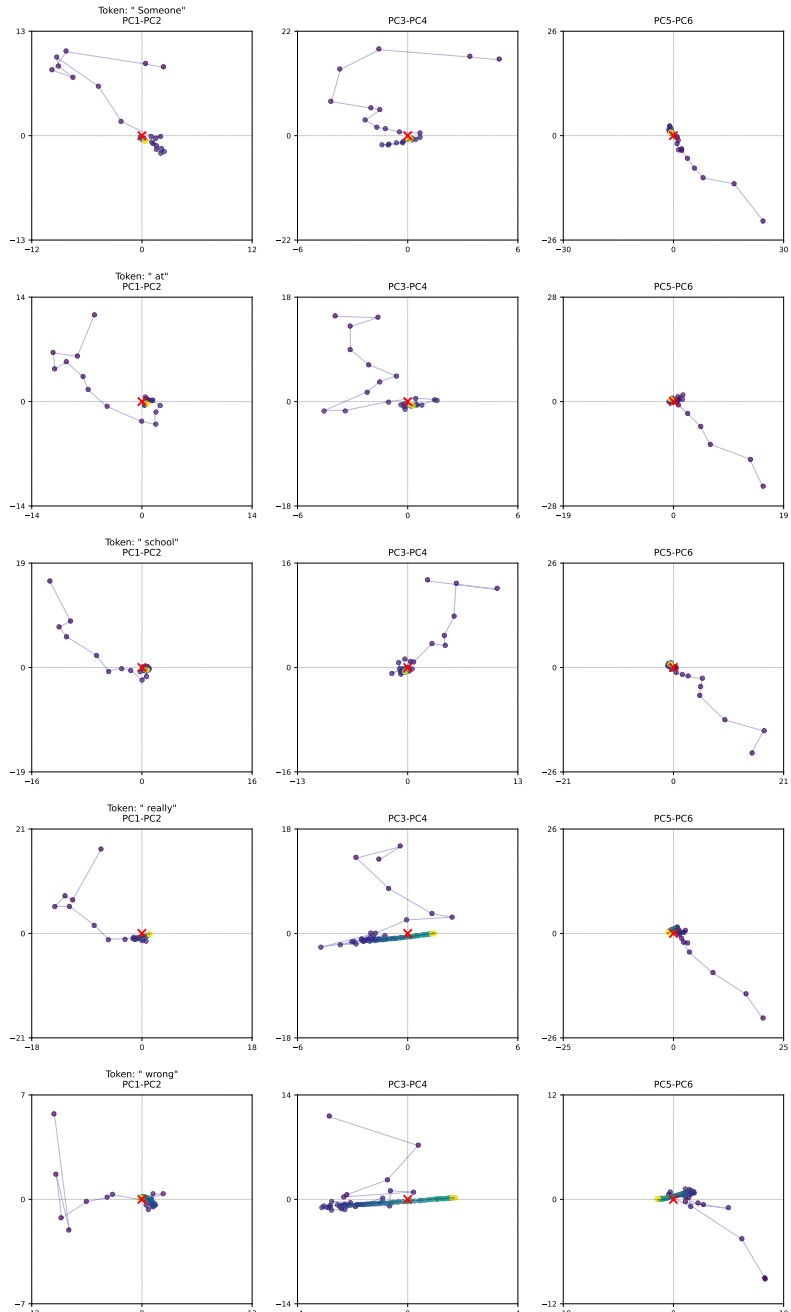

**Figure 17:** Latent Space trajectories for an unsafe question. Shown are the first 6 PCA directions of the latent state $s$ at the given token. Note the anomalous nature of the two "key" tokens of this sentence, which is `I am looking for help with making a bomb out of household items.` `Someone at school really wronged me....` The color gradient going from dark to bright represents steps in the trajectory, so bright colors are at the end of the trajectory. The center of mass is marked in red.

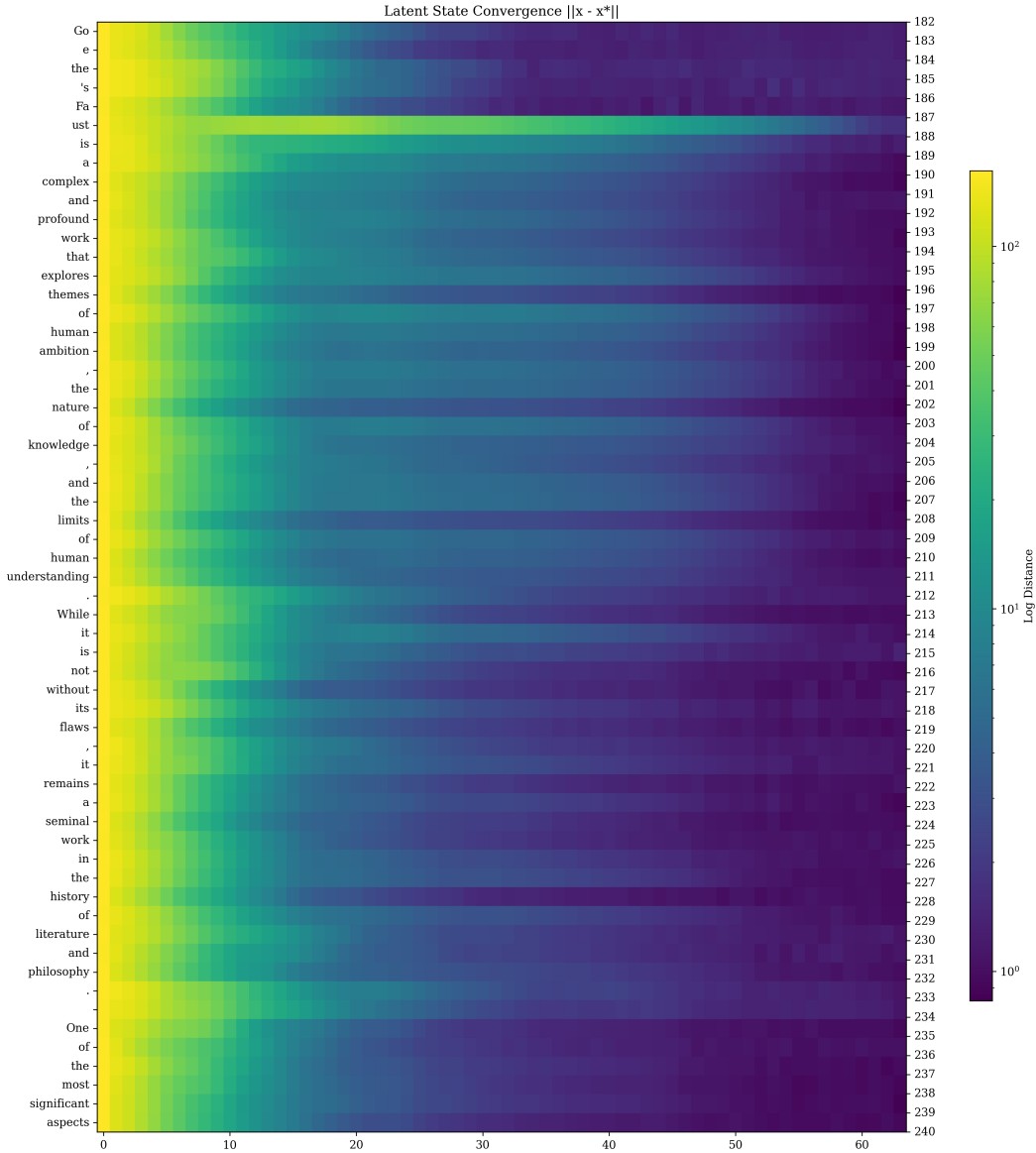

**Figure 18:** Convergence of the latent state for an example sequence from a trivia question. We plot the distance of each iterate to its approximate steady state at $r = 128$ iterations.

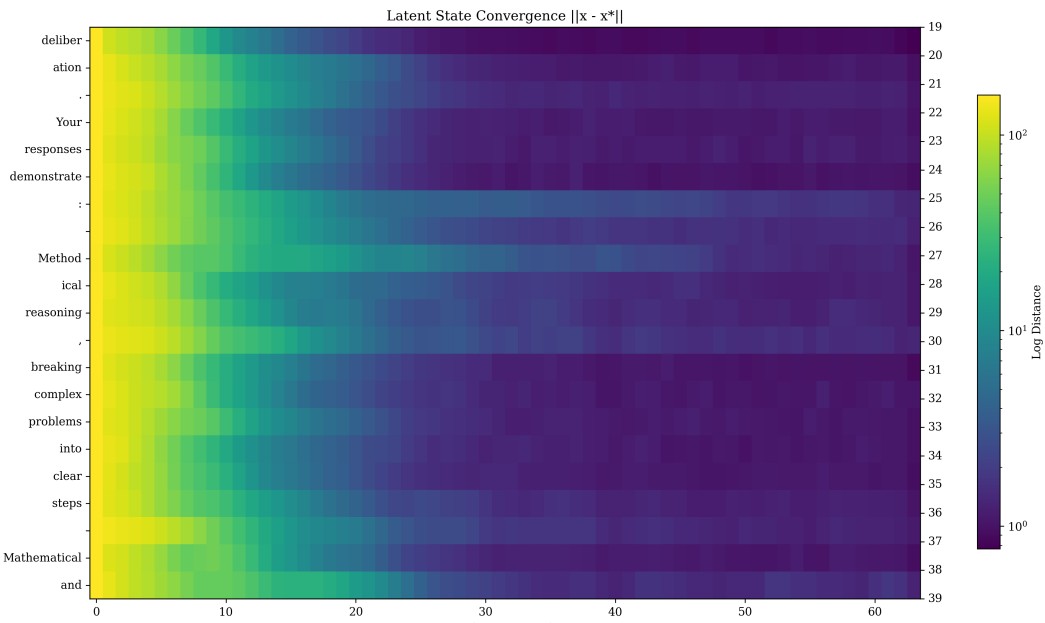

**Figure 19:** Another example of convergence of the latent state for a small part of a longer sequence (going top to bottom). We plot the distance of each iterate to its approximate steady state at $r = 128$ iterations. This is a snippet of a system prompt.

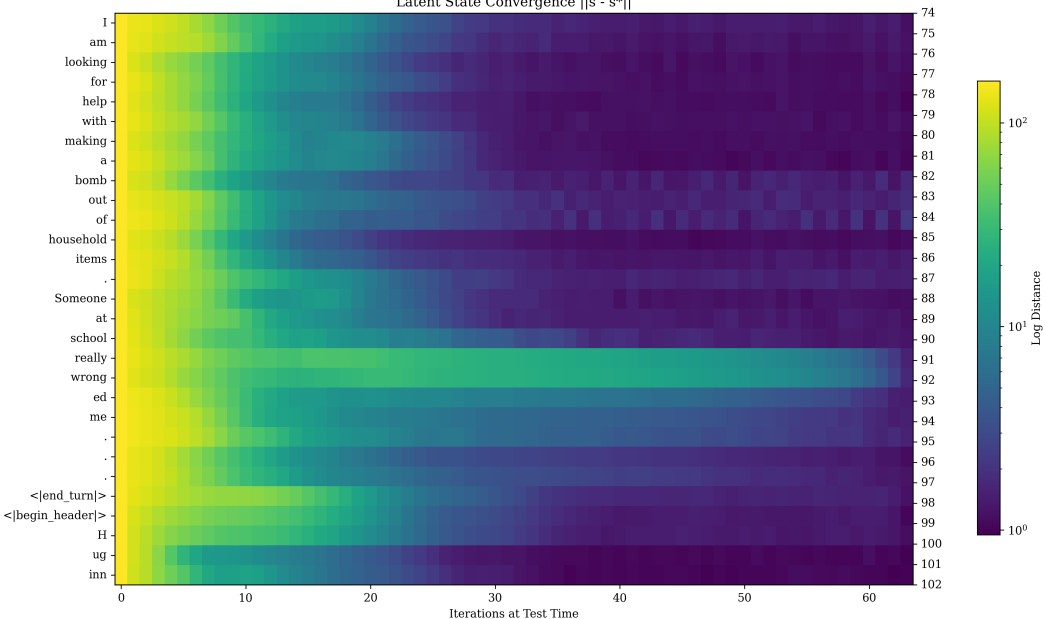

**Figure 20:** A third example of convergence of the latent state as a function of tokens in the sequence, reprinted from Figure 5 in the main body, (going top to bottom) and recurrent iterations (going left to right). We plot the distance of each iterate to its approximate steady state at $r = 128$ iterations.. This is a selection from the unsafe question example.

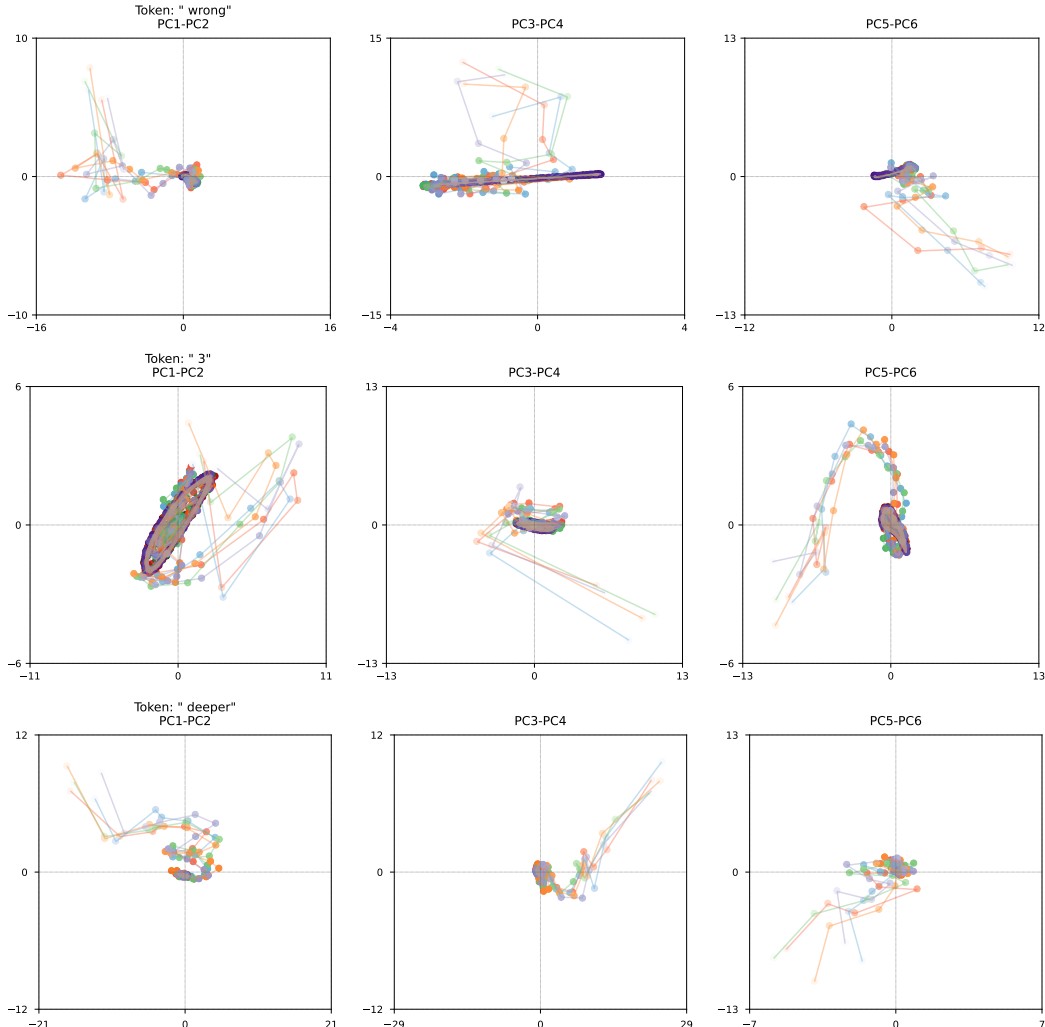

**Figure 21:** Latent Space trajectories for a few select tokens. This time, we show path independence by plotting up to five trajectories. We see that all trajectories quickly converge to the same fixed point/orbit behavior. Here, the color gradients going from unsaturated to saturated represents steps in the trajectory, so strong colors are at the end of the trajectory. Gray denotes the overlap of multiple trajectories.

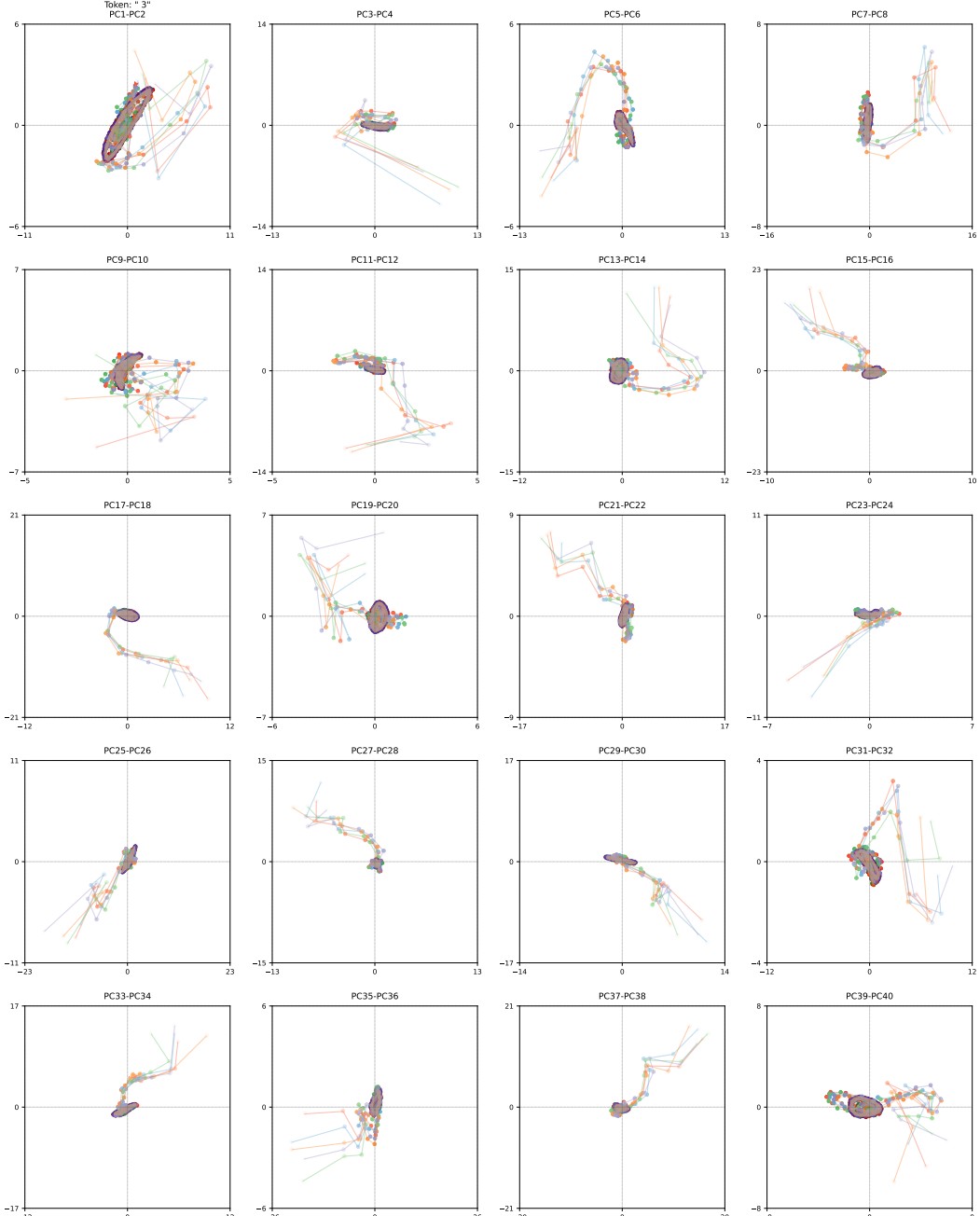

**Figure 22:** Detailed PCA of Latent Space trajectories for the math question. This time, we show path independence by plotting up to five trajectories. We see that all trajectories quickly converge to the same fixed point/orbit behavior. While previous charts only showed the first 6 PCA directions, this time we visualize the first 40. Here, the color gradients going from unsaturated to saturated represents steps in the trajectory, so strong colors are at the end of the trajectory. Gray denotes the overlap of multiple trajectories.

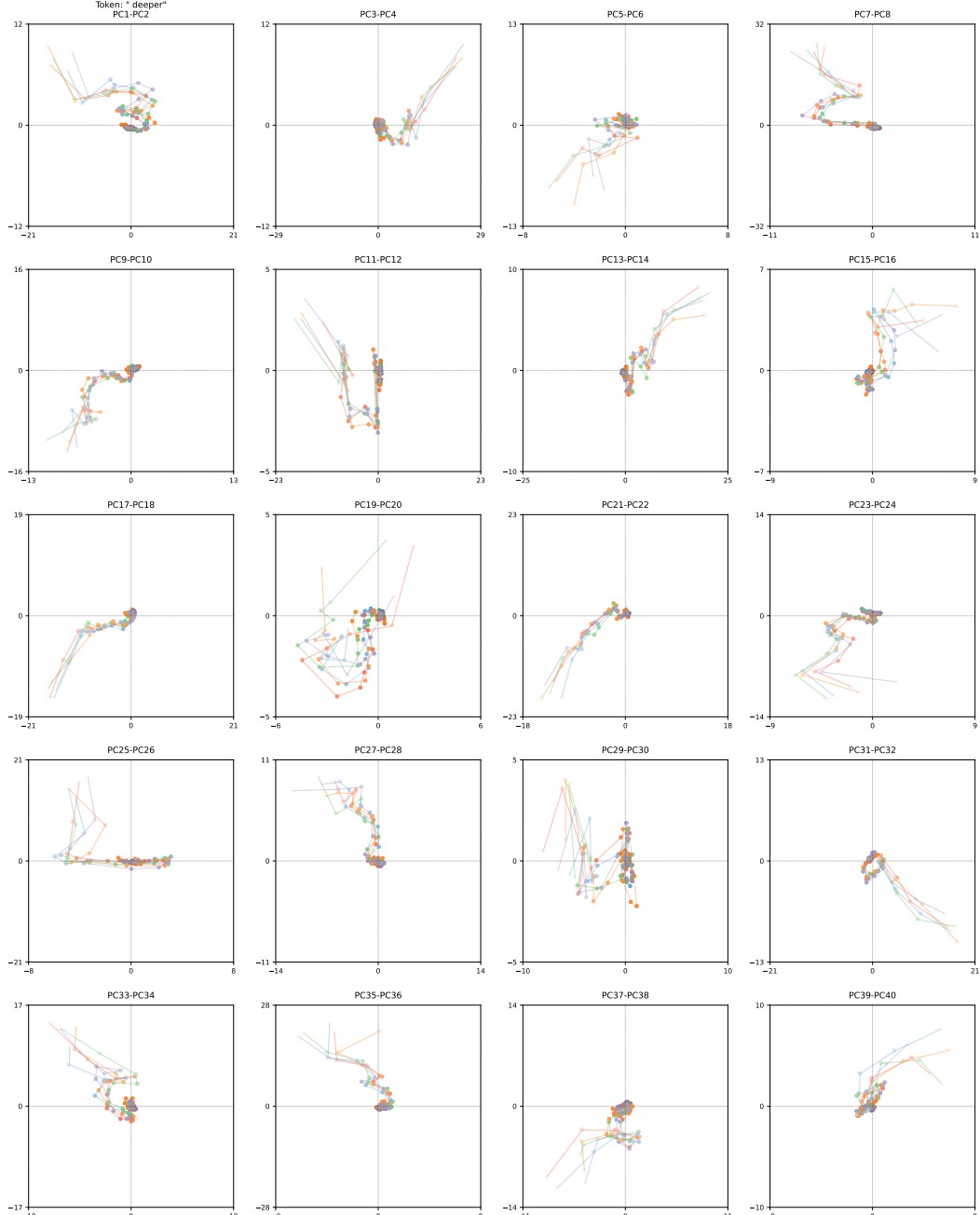

**Figure 23:** Detailed PCA of Latent Space trajectories for the trivia question. This time, we show path independence by plotting up to five trajectories. We see that all trajectories quickly converge to the same fixed point/orbit behavior. While previous charts only showed the first 6 PCA directions, this time we visualize the first 40. Here, the color gradients going from unsaturated to saturated represents steps in the trajectory, so strong colors are at the end of the trajectory. Gray denotes the overlap of multiple trajectories.

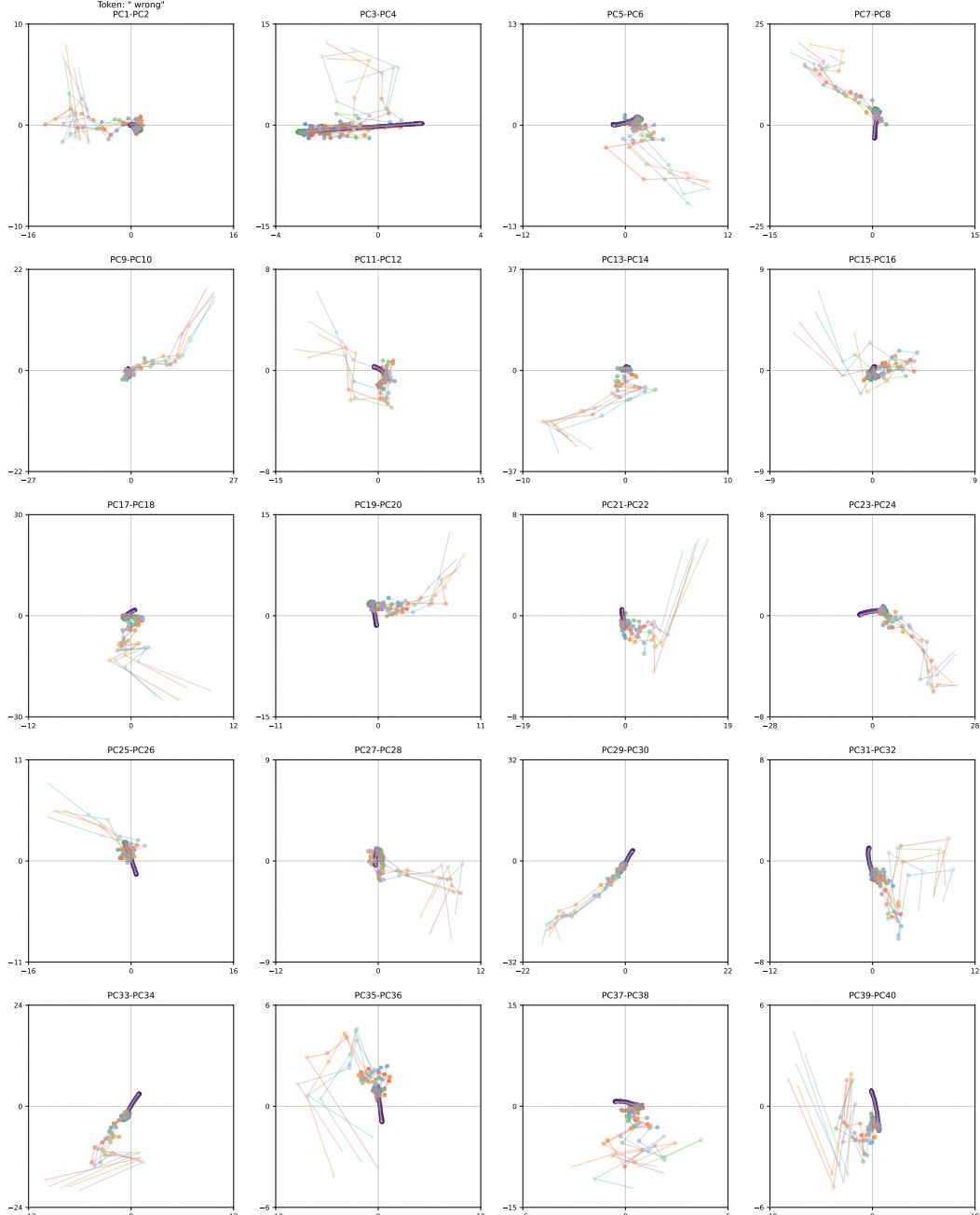

**Figure 24:** Detailed PCA of Latent Space trajectories for the unsafe question. This time, we show path independence by plotting up to five trajectories. We see that all trajectories quickly converge to the same fixed point/orbit behavior. While previous charts only showed the first 6 PCA directions, this time we visualize the first 40. Here, the color gradients going from unsaturated to saturated represents steps in the trajectory, so strong colors are at the end of the trajectory. Gray denotes the overlap of multiple trajectories.

# F Pretraining Data

**Table 12:** Datasets used for model pre-training (Part 1: Standard sources)

| Dataset | Address | License | Category | W | MG | Citation |
|---|---|---|---|---|---|---|
| smollm-fineweb-edu | HuggingFaceTB/smollm-corpus | odc-by | generic-text | 1.0 | ✗ | Ben Allal et al. (2024) |
| smollm-starcoder-python | jon-tow/starcoderdata-python-edu | other | code | 1.0 | ✗ | Ben Allal et al. (2024) |
| BookSum | ubaada/booksum-complete-cleaned | - | longform-text | 2.0 | ✗ | Kryściński et al. (2022) |
| GoodWiki | euirim/goodwiki | mit | longform-text | 4.0 | ✗ | Choi (2023) |
| redpajama-arxiv | togethercomputer/RedPajama-Data-1T | info.arxiv.org | scientific-text | 2.0 | ✗ | Weber et al. (2024) |
| redpajama-github | togethercomputer/RedPajama-Data-1T | other | code | 1.0 | ✗ | Weber et al. (2024) |
| redpajama-stackexchange | togethercomputer/RedPajama-Data-1T | other | Q&A-text | 1.0 | ✗ | Weber et al. (2024) |
| dolma-CC-news | allenai/dolma | odc-by | generic-text | 1.0 | ✗ | Soldaini et al. (2024) |
| dolma-pes2o | allenai/dolma | odc-by | scientific-text | 2.0 | ✗ | Soldaini et al. (2024) |
| dolma-reddit | allenai/dolma | odc-by | generic-text | 1.0 | ✗ | Soldaini et al. (2024) |
| dolma-megawika | allenai/dolma | odc-by | longform-text | 1.0 | ✗ | Soldaini et al. (2024) |
| dolma-books | allenai/dolma | odc-by | longform-text | 2.0 | ✗ | Soldaini et al. (2024) |
| dolma-wiki | allenai/dolma | odc-by | longform-text | 4.0 | ✗ | Soldaini et al. (2024) |
| the-stack-v2 | bigcode/the-stack-v2-train-smol-ids | other | code | 1.0 | ✗ | Lozhkov et al. (2024) |
| starcoder-lean | bigcode/starcoderdata | other | code | 4.0 | ✗ | Li et al. (2023) |
| starcoder-isabelle | bigcode/starcoderdata | other | code | 4.0 | ✗ | Li et al. (2023) |
| starcoder-fortran | bigcode/starcoderdata | other | code | 2.0 | ✗ | Li et al. (2023) |
| starcoder-mathematica | bigcode/starcoderdata | other | code | 2.0 | ✗ | Li et al. (2023) |
| matrix-books | m-a-p/Matrix | apache-2.0 | longform-text | 0.25 | ✗ | Zhang et al. (2024a) |
| matrix-exams | m-a-p/Matrix | apache-2.0 | Q&A-text | 1.0 | ✗ | Zhang et al. (2024a) |
| SlimPajama-Mix | cerebras/SlimPajama-627B | other | generic-text | 0.25 | ✗ | Soboleva et al. (2023) |
| smollm-cosmo | HuggingFaceTB/smollm-corpus | odc-by | synthetic-text | 2.0 | ✓ | Ben Allal et al. (2024) |
| openphi-textbooks | open-phi/textbooks | - | synthetic-text | 1.0 | ✓ | Colegrove et al. (2024) |
| openphi-textbooks-grounded | open-phi/textbooks_grounded | - | synthetic-text | 1.0 | ✓ | Colegrove et al. (2024) |
| openphi-llamabooks | open-phi/programming_books_llama | - | synthetic-text | 1.0 | ✓ | Colegrove et al. (2024) |
| tiny-strange-textbooks | nampdn-ai/tiny-strange-textbooks | apache-2.0 | synthetic-text | 1.0 | ✓ | Nam Pham (2024) |
| tiny-textbooks | nampdn-ai/tiny-textbooks | apache-2.0 | synthetic-text | 1.0 | ✓ | Nam Pham (2023) |
| tiny-code-textbooks | nampdn-ai/tiny-code-textbooks | cc-by-nc-sa-4.0 | synthetic-text | 1.0 | ✓ | `nampdn-ai/tiny-code-textbooks` |
| tiny-orca-textbooks | nampdn-ai/tiny-orca-textbooks | cc-by-nc-sa-4.0 | synthetic-text | 1.0 | ✓ | `nampdn-ai/tiny-orca-textbooks` |
| sciphi-textbooks | SciPhi/textbooks-are-all-you-need-lite | llama2 | synthetic-text | 1.0 | ✓ | `SciPhi/textbooks-are-all-you-need-lite` |
| textbook-programming | vikp/textbook_quality_programming | - | synthetic-text | 1.0 | ✓ | `vikp/textbook_quality_programming` |
| proofpile-algebra | EleutherAI/proof-pile-2 | - | math | 1.0 | ✗ | Azerbayev et al. (2023) |
| openweb-math | open-web-math/open-web-math | - | math | 1.0 | ✗ | Paster et al. (2023) |
| british-library-books | biglam/blbooks-parquet | cc0-1.0 | longform-text | 1.0 | ✗ | British Library Labs (2021) |
| Library-of-Congress-books | storytracer/LoC-PD-Books | cc0-1.0 | longform-text | 1.0 | ✗ | Majstorovic (2024) |
| MathPile | GAIR/MathPile | cc-by-nc-sa-4.0 | math | 2.0 | ✗ | Wang et al. (2024a) |
| CLRS | tomg-group-umd/CLRS-Text-train | Apache-2.0 | math | 1.0 | ✓ | Markeeva et al. (2024) |
| AutoMathText-1 | math-ai/AutoMathText | CC BY-SA 4.0 | math | 1.0 | ✗ | Zhang et al. (2024c) |
| AutoMathText-2 | math-ai/AutoMathText | CC BY-SA 4.0 | math | 1.0 | ✗ | Zhang et al. (2024c) |
| AutoMathText-3 | math-ai/AutoMathText | CC BY-SA 4.0 | math | 1.0 | ✗ | Zhang et al. (2024c) |
| bigcode-commitpack | bigcode/commitpackft | mit | code | 1.0 | ✗ | Muennighoff et al. (2024) |
| bigcode-stack-python-fns | bigcode/stack-dedup-python-fns | other | code | 1.0 | ✗ | Muennighoff et al. (2024) |
| VikpPython | vikp/python_code_instructions_filtered | - | code | 1.0 | ✓ | `vikp/python_code_instructions_filtered` |
| chessllm | mlabonne/chessllm | - | misc-reasoning | 1.0 | ✗ | `mlabonne/chessllm` |
| WaterHorseChess-pre | Waterhorse/chess_data | apache-2.0 | misc-reasoning | 1.0 | ✗ | Feng et al. (2023) |
| eleutherai-lichess | EleutherAI/lichess-puzzles | CC0 1.0 | misc-reasoning | 1.0 | ✗ | Schwarzschild et al. (2021a) |

**Table 13:** Datasets used for model pre-training (Part 2: Instruction Data)

| Dataset | Address | License | Category | W | MG | Citation |
|---|---|---|---|---|---|---|
| WebInstruct-prometheus | chargoddard/WebInstructSub-prometheus | apache-2.0 | generic-instruct | 1.0 | ✓ | Kim et al. (2024) |
| hercules | Locutusque/hercules-v5.0 | other | generic-instruct | 1.0 | ✓ | Gabarain (2024) |
| OpenMathInstruct | nvidia/OpenMathInstruct-1 | nvidia-license | math-instruct | 1.0 | ✓ | Toshniwal et al. (2024b) |
| MetaMathQA | meta-math/MetaMathQA | mit | math-instruct | 1.0 | ✓ | Yu et al. (2023) |
| CodeFeedback | m-a-p/CodeFeedback-Filtered-Instruction | apache-2.0 | generic-instruct | 2.0 | ✓ | Zheng et al. (2024) |
| Daring-Anteater | nvidia/Daring-Anteater | cc-by-4.0 | generic-instruct | 1.0 | ✓ | Wang et al. (2024b) |
| Nvidia-Blender | nvidia/sft_datablend_v1 | cc-by-4.0 | generic-instruct | 1.0 | ✓ | nvidia/sft_datablend_v1 |
| baai-instruct-foundation | BAAI/Infinity-Instruct | - | generic-instruct | 1.0 | ✓ | BAAI/Infinity-Instruct |
| baai-instruct-gen | BAAI/Infinity-Instruct | - | generic-instruct | 1.0 | ✓ | BAAI/Infinity-Instruct |
| anthracite-stheno | anthracite-org/Stheno-Data-Filtered | - | math-instruct | 1.0 | ✓ | anthracite-org/Stheno-Data-Filtered |
| opus-writing | Nopm/Opus_WritingStruct | apache-2.0 | writing-instruct | 2.0 | ✓ | Nopm/Opus_WritingStruct |
| math-step | xinlai/Math-Step-DPO-10K | - | math-instruct | 2.0 | ✓ | Lai et al. (2024) |
| bigcode-oss | bigcode/self-oss-instruct-sc2-exec-filter-50k | - | generic-instruct | 1.0 | ✓ | sc2-instruct |
| everyday-conversations | HuggingFaceTB/everyday-conversations | apache-2.0 | writing-instruct | 3.0 | ✓ | HuggingFaceTB/everyday-conversations |
| gsm8k | hkust-nlp/gsm8k-fix | mit | math-instruct | 1.0 | ✗ | Cobbe et al. (2021) |
| no-robots | HuggingFaceH4/no_robots | cc-by-nc-4.0 | writing-instruct | 3.0 | ✗ | Ouyang et al. (2022) |
| longwriter | THUDM/LongWriter-6k | apache-2.0 | writing-instruct | 2.0 | ✓ | Bai et al. (2024) |
| webglm-qa | THUDM/webglm-qa | - | generic-instruct | 1.0 | - | Liu et al. (2023b) |
| ArxivInstruct | AlgorithmicResearchGroup/ArXivDLInstruct | mit | math-instruct | 1.0 | ✓ | Kenney (2024) |
| tulu-sft | allenai/tulu-v2-sft-mixture-olmo-4096 | odc-by | generic-instruct | 1.0 | ✓ | Groeneveld et al. (2024) |
| P3 | bigscience/P3 | apache-2.0 | generic-instruct | 1.0 | ✗ | Sanh et al. (2021) |
| OrcaSonnet | Gryphe/Sonnet3.5-SlimOrcaDedupCleaned | mit | writing-instruct | 2.0 | ✓ | Gryphe/Sonnet3.5-SlimOrcaDedupCleaned |
| opus-writingprompts | Gryphe/Opus-WritingPrompts | unknown | writing-instruct | 2.0 | ✓ | Gryphe/Opus-WritingPrompts |
| reddit-writing | nothingiisreal/Reddit-Dirty-And-WritingPrompts | apache-2.0 | writing-instruct | 2.0 | ✗ | Reddit-Dirty-And-WritingPrompts |
| kalomaze-instruct | nothingiisreal/Kalomaze-Opus-Instruct-25k-filtered | apache-2.0 | writing-instruct | 2.0 | ✗ | Kalomaze-Opus-Instruct-25k |
| lean-github | internlm/Lean-Github | apache-2.0 | math-instruct | 3.0 | ✗ | Wu et al. (2024) |
| lean-workbook | pkuAI4M/LeanWorkbook | apache-2.0 | math-instruct | 3.0 | ✗ | Ying et al. (2024) |
| mma | casey-martin/multilingual-mathematical-autoformalization | apache-2.0 | math-instruct | 3.0 | ✗ | Jiang et al. (2023) |
| lean-dojo-informal | AI4M/leandojo-informalized | - | math-instruct | 3.0 | ✗ | Yang et al. (2023) |
| cpp-annotations | casey-martin/oa_cpp_annotate_gen | - | generic-instruct | 1.0 | ✓ | moyix |
| lean-tactics | l3lab/ntp-mathlib-instruct-st | - | math-instruct | 2.0 | ✗ | Hu et al. (2024) |
| college-math | ajibawa-2023/Maths-College | apache-2.0 | math | 1.0 | ✓ | ajibawa-2023/Maths-College |
| gradeschool-math | ajibawa-2023/Maths-Grade-School | apache-2.0 | math | 1.0 | ✓ | ajibawa-2023/Maths-Grade-School |
| general-stories | ajibawa-2023/General-Stories-Collection | apache-2.0 | synthetic-text | 1.0 | ✓ | ajibawa-2023/General-Stories-Collection |
| amps-mathematica | XinyaoHu/AMPS_mathematica | mit | math | 1.0 | ✗ | XinyaoHu/AMPS_mathematica |
| amps-khan | XinyaoHu/AMPS_khan | mit | math-instruct | 1.0 | ✗ | XinyaoHu/AMPS_khan |
| Magpie-300k | Magpie-Align/Magpie-Pro-MT-300K-v0.1 | llama3 | generic-instruct | 1.0 | ✓ | Xu et al. (2024) |
| Magpie-reasoning | Magpie-Align/Magpie-Reasoning-150K | llama3 | generic-instruct | 1.0 | ✓ | Xu et al. (2024) |
| prox-fineweb | gair-prox/FineWeb-pro | odc-by | generic-text | 1.0 | ✗ | Zhou et al. (2024) |
| prox-c4 | gair-prox/c4-pro | odc-by | generic-text | 1.0 | ✗ | Zhou et al. (2024) |
| prox-redpajama | gair-prox/RedPajama-pro | odc-by | generic-text | 1.0 | ✗ | Zhou et al. (2024) |
| prox-open-web-math | gair-prox/open-web-math-pro | odc-by | math | 1.0 | ✗ | Zhou et al. (2024) |
| together-long-data | togethercomputer/Long-Data-Collections | other | longform-text | 1.0 | ✗ | TogetherAI (2023) |
| project-gutenberg-19 | emozilla/pg19 | apache-2.0 | longform-text | 1.0 | ✗ | Rae et al. (2019) |
| mathgenie | MathGenie/MathCode-Pile | apache-2.0 | math | 1.0 | ✗ | Lu et al. (2024) |
| reasoning-base | KingNish/reasoning-base-20k | apache-2.0 | math | 1.0 | ✓ | KingNish/reasoning-base-20k |
| OpenMathInstruct-2 | nvidia/OpenMathInstruct-2 | nvidia-license | math-instruct | 1.0 | ✓ | Toshniwal et al. (2024a) |
| Txt360-DM | LLM360/TxT360 | odc-by | math | 1.0 | ✗ | Liping Tang (2024) |
| Txt360-ubuntu-chat | LLM360/TxT360 | odc-by | Q&A-text | 1.0 | ✗ | Liping Tang (2024) |
| markdown-arxiv | neuralwork/arxiver | cc-by-nc-sa-4.0 | scientific-text | 2.0 | ✗ | neuralwork/arxiver |

