# OpenReview forum: "Scaling up Test-Time Compute with Latent Reasoning: A Recurrent Depth Approach"
_NeurIPS.cc/2025/Conference — NeurIPS 2025 spotlight_

### Official Review · Reviewer_6TRY · 2025-06-30

**Clarity:** 2
**Significance:** 3
**Originality:** 3
**Rating:** 5
**Confidence:** 4

**Summary:**

This paper designs a recurrent transformer architecture and evaluates it on language modeling tasks. The model is trained with a varying number of recurrences. This enables adaptive test-time compute by applying the recurrent block a varying number of times. The authors compare to a nonrecurrent model of the same architecture. The recurrent model with high test time compute performs slightly better. In addition, the authors provide an analysis of the latent space trajectories. "Overall, we believe this may be the largest language model training run to completion in terms of number of devices used in parallel on an AMD cluster, as of time of writing"

**Questions:**

Q1. Is it fair to compare the recurrent model, when evaluated at R=32, to a 32B parameter model in terms of training Flops, or 50B for test flops? Or are there other metrics that are useful for comparing the model to baselines "of equal ability" ? Maybe it would be useful to show the number of flops for each model in tables such as Table 1 and Table 4.

Q2. Section 5: the KL divergence between two successive steps is used for adaptive compute. What exactly is the KL divergence between? The latent state? Figure 4 is really confusing... what is Continuous CoT? what is Default? These labels are never described.

Q3. It's unclear to me whether there's any insight in the analysis in section 6. Can the same analysis be applied to nonrecurrent models, but looking at how the latent state between layers evolves over time? Have such analyses been done, and do the results here differ? The paper does mention that similar multi-dimensional orbits have been observed in fixed-depth transformers for arithmetic tasks.

**Ethical Concerns:**

["NO or VERY MINOR ethics concerns only"]

**Final Justification:**

I would like to give this paper a higher score, because the model trained is really impressive and interesting, but I can't help but still think that the results are presented a bit misleadingly. That said, I think it should be accepted regardless.

**Paper Formatting Concerns:**

Several figures are too small to read (see above)

**Quality:**

3

**Strengths And Weaknesses:**

The main strength of this paper is that the authors train a really large model, with a novel architecture to see how good the results are.
The biggest value in this paper is Table 4, specifically the first three rows of it. Here, we can see how good the recurrent model is compared to the non-recurrent baseline. This is very scientifically useful.

The main weakness of the paper is that the "message" of the paper is very confused. The authors are trying to argue that the recurrent approach could have advantages, but the results of this large scale experiment seem to indicate that the current recurrent model approach is not worth. I think the paper would be much stronger with a clearer message. It feels like the authors threw a lot of random things in the paper to try to make it better, but these all feel like distractions compared to the main point of this paper, which is experimenting with a huge recurrent model training run.

I think of this paper as a "negative result" for recurrent architectures, that is still worth being published. It's a huge experiment and this work should clearly be published. However, I think the framing of the paper obfuscates the most valuable lessons from this paper. I think the authors probably threw in a lot of other analyses to make the paper seem stronger, but I would rather have it more clearly present the negative result to make it easier for others to build on it.

Table 4: this table shows that compared to a fixed-depth baseline, the recurrent model only performs a bit better when using R=32 recurrences. The paper states that the recurrent model "consumes flops close to what a 32B parameter transformer would consume during pretraining" and the R=32 test time compute is comparable in flops to a 50B parameter fixed depth. So this approach seems really not worth it — almost 10x parameters to the baseline, and hardly higher in performance.

The remaining comparisons, such as table 1, I don't find to be very useful, given how many flops the model consumes in comparison to the other models.

## Strengths
- The paper trains a really large model. "Overall, we believe this may be the largest language model training run to completion in terms of number of devices used in parallel on an AMD cluster, as of time of writing" — wow!
- The recurrent model seems well-designed and is described well in the paper.
- The authors provide a ton of various analyses of the model, and a thorough appendix with their training methodology.

## Weaknesses
- Many of the figures are far too small to read when the paper is printed out. (4, 5, 6, and many in the appendix) — need to make sure the axes and labels can be read.
- Many of the sections of the paper are not very enlightening. Table 4 is the main payday.
  - Section 4.4 is just about ensembling to improve metrics.
  - Section 6 tries to analyze the mechanisms in the recurrent model, but it's unclear whether the analysis uncovers anything that's not already present in fixed-depth models (see questions section)
  - Figure 4 is very poorly explained (see questions section)

---

> ### Author Rebuttal · Authors · 2025-07-31
>
> Thank you for your extensive feedback and detailed reading of our paper. We’re glad that you consider the result worth being published. We have tried hard to pack as much scientific value into our paper as we can, and to let the results stand for themselves. If future research ultimately concludes that our paper is simply evidence that the benefits of recurrent depth are not worth their costs, then we will still consider our endeavor, and this manuscript, a success.
>
> ---
> &nbsp;
>
> > I think of this paper as a "negative result" for recurrent architectures, that is still worth being published.
>
> While we don’t want to disabuse this reading of the paper, and hope that we have presented all results as neutrally as possible, we do want to argue that Table 4 shows (to us) the key wins of recurrent depth in our study and where to expect them in practice. Namely, in Table 4, the recurrent-depth model shows drastic gains in reasoning performance on GSM8k (a factor of 5x over the non-recurrent baseline!) and reasonable gains on deducing the correct answer on HellaSwag (37% baseline to 48% recurrent model). However, it is true that little gain is realized on knowledge-heavy tasks such as OBQA, PiQA or SciQ over the non-recurrent baseline.
>
> We think this accurately represents the potential of recurrent depth models for “reasoning” (see also the discussion in the motivation section). It is indeed a suboptimal use of compute to try and use recurrence to improve performance if the primary focus is on tasks requiring retrieval of associative relations about the world and/or deductions from the large amount of knowledge stored within the trainable parameters of the model. However, both our intuitions and our empirical results suggest that in domains where, given the same knowledge, more _computation_ can be expected to yield more accurate results, recurrence can unlock notable improvements. This is also shown in Fig. 1b. On a number of tasks such as SciQ, running the recurrence for 1 or 4 steps provides most of the gains, likely meaning that the model either retrieves the relevant fact, or does not know it. On the other hand, for tasks like HumanEval or Mastermind (the reasoning game) while the model performs at random chance with limited compute, additional recurrent computation again provides substantial improvements.
>
> ---
> &nbsp;
>
> > figures are far too small to read when the paper is printed out
>
> Thank you for bringing this to our attention! We will certainly address this in the updated version of the paper.
>
> ---
> &nbsp;
>
> > Section 4.4 is just about ensembling
>
> Just to be safe, we want to highlight that the approach in Section 4 is using offline checkpoint merging, also sometimes described as “latest weight averaging” (e.g. Kaddour et al. 2022), not an ensemble, so the final result is a single model that can be used for downstream applications as normal, as such we consider this part of our pretraining pipeline.
>
> ---
> &nbsp;
>
> Regarding your questions:
>
> > Q1. Is it fair to compare the recurrent model, when evaluated at R=32, to a 32B parameter model in terms of training Flops, or 50B for test flops?
>
> We agree that a straightforward comparison is not so easy, so we have opted to just present the data regarding training flops and inference flops. We also note that the model is generally faster to train/infer than flop-equivalent fixed-depth transformers, given that, due to its smaller size, costs for communication during training, and model memory during inference, are both improved.  We also note that the comparison would have to be to a 32B parameter model trained for the same number of tokens. Maybe even better, it would be a comparison between two models trained on the same compute cluster for the same wallclock time with similar engineering. In our table we can only point to the Pythia series, trained on fewer tokens, and the Olmo series trained on significantly more tokens.
>
> We also want to re-iterate that, for the first (and the only) recurrent model trained at this scale as a large language model, the results are encouraging in comparison to these open source models building on training recipes for fixed-depth models that are by now much more established.
>
> ---
> &nbsp;
>
> > it would be useful to show the number of flops for each model in tables such as Table 1 and Table 4.
>
> This is a good point, we’ll add pretraining and inference flops to these tables.
>
> ---
> &nbsp;
>
> > Q2. Section 5: the KL divergence between two successive steps is used for adaptive compute. What exactly is the KL divergence between?
>
> The metric described in section 5 is the KL divergence between the probability distribution over the next token at step r-1 to the probability distribution over the next token at r. This is just one option on how to exit that we considered natural when writing the paper, and using, for example, the Euclidean distance between the latent states is also a successful strategy. We’ll clarify this part in our revised version and can provide more detailed variants of all of the experiments on decoding strategies in the appendix.
>
> ---
>
> > what is Continuous CoT?
>
> Continuous CoT is an inference variant described in Appendix Section D.6, namely warm-starting the latent state at each generated token with the last latent state from the previous token. We moved this section into the appendix at the last minute. We’ll remove this variant from Fig.4 in the main body, and show this only in the appendix in the future. Sorry for the confusion!.
>
> ---
> &nbsp;
>
> > Q3. It's unclear to me whether there's any insight in the analysis in section 6. Can the same analysis be applied to nonrecurrent models,
>
> Actually, this kind of analysis is already a bit special to the recurrent model. Due to the use of a single non-linear operator, the recurrent block, all latent states naturally lie in the same “space” and can be analyzed as an iterative series. For normal models, there is some pressure due to the residual connections to keep representations similar from layer to layer, but there is no requirement. Every layer in a transformer is initialized randomly and at minimum, arbitrarily rotates the features of the model. There is no requirement for e.g. the n-th feature of the latent state to correspond to the same feature across layers of the model like it does for the recurrent-depth model. Only in a few specialized examples, like a model trained only for modular addition (as referenced) were similar patterns like the orbits shown.
>
> ---
> &nbsp;
>
> Overall, we thank you again for your detailed feedback and believe it has helped improve the presentation and outlook of our paper. In light of our results showcasing how certain “computation” centric tasks like math, logical reasoning, and code generation benefit from the application of recurrent compute, we think that our research does demonstrate the promise of the recurrent approach and warrants further exploration and development effort.
>
> However, we share the reviewer’s perspective that our results also suggest that for some usecases where the raw capacity for knowledge storage within the parameters of the model is the primary determiner of performance, a recurrent approach might be suboptimal. Ultimately, we’re optimistic that our study of recurrent depth large language models at scale will be a valuable addition to the literature for both researchers and practitioners interested in pushing the limits of efficiency and capability across a broad spectrum of tasks and deployment usecases.
>
>
>
>
> &nbsp;
>
> &nbsp;
>
>
> *References*:
>
> Kaddour, Jean. "Stop wasting my time! saving days of imagenet and bert training with latest weight averaging." arXiv preprint arXiv:2209.14981 (2022).

---

> > ### Comment · Reviewer_6TRY · 2025-08-01
> >
> > Thank you for taking the time to answer my questions. That said, while I am usually convinced by author rebuttals, I am not as convinced by this one.
> >
> > The main evidence presented that the approach is not a "negative result" is table 4. however, given the model has high training and inference compute costs of the approach (32B training equivalent, 50B inference equivalent!), the fact that the model outperforms the 3.5B non-recurrent baseline does not seem very notable. I agree that the best comparison would be a 32B nonrecurrent model trained on the same data and # tokens, but understand that this is not feasible in terms of computational resources. But the appropriate conclusion from what is presented in the paper seems to be sometehing like "recurrent models have some advantages, but are very computationally expensive, so it is hard to have a good comparison to know whether they are ultimately worth it when designing new LLM architectures."
> >
> > I would like to give this paper a higher score, because the model trained is really impressive and interesting, but I can't help but still think that the results are presented a bit misleadingly. That said, I think it should be accepted regardless.

---

> > > ### Author Response · Authors · 2025-08-05
> > >
> > > Thank you for the clarification! As mentioned, we don't want to push back on this reading of the results, and we're glad you find them helpful either way. We'll work on updating our writing to clarify that the missing gains on, e.g. knowledge tasks, are a tangible limitation of this approach.
> > >
> > > Regarding the comparison though, we have recently been made aware that intermediate checkpoints are available for the Olmo-2-32b model. This allows us to evaluate a checkpoint, namely `allenai/OLMo-2-0325-32B,revision=stage1-step100000-tokens839B`, which is trained on different data, but at least in a similar range of FLOPs (being a 32B model) and with the same number of tokens (~800B). The Olmo-2-32b fixed-depth transformer is certainly stronger on knowledge tasks, but interestingly, on all of HumanEval (coding), GSM8k (grade-school math) and Mastermind (a reasoning game) - the 32b checkpoint models lags behind the results from the recurrent-depth model:
> > > | Model | Tokens (B) | Parameters (B) | HumanEval | GSM8k | Mastermind  |
> > > |-------|------------|----------------|---------------------------|-------|----------------------|
> > > | Olmo-2-32b-checkpoint | 839 | 32 | 0.1768 | 0.1228 | 0.5402 |
> > > | Recurrent-Depth Model | 800 | 3.5 | **0.2317** | **0.3813** | **0.58** |
> > >
> > > Of course, this is different data (namely the olmo pretraining dataset), we're only noting it, because we have recently completed this comparison, which we'll include in the next version of our draft, to complement our existing comparisons.

---

> > > > ### Comment · Reviewer_6TRY · 2025-08-05
> > > >
> > > > That's a great find and super interesting, definitely an enlightening comparison to have!

---

### Official Review · Reviewer_TR9E · 2025-07-01

**Clarity:** 3
**Significance:** 3
**Originality:** 3
**Rating:** 5
**Confidence:** 2

**Summary:**

This paper introduces a novel language model architecture that scales test-time computation through "latent reasoning" by iterating a shared, recurrent block of layers. This recurrent depth approach is presented as an alternative to mainstream methods. The authors demonstrate that their 3.5B parameter model significantly improves on reasoning-intensive tasks, such as math and coding, as more computational steps (recurrences) are applied at test time. A key contribution is showing that this architecture also inherently enables several inference optimizations, including adaptive compute, KV-cache sharing, and speculative decoding.

**Questions:**

1. Could you provide a more direct comparison of the inference FLOPs? For example, for GSM8k, how many FLOPs does your model at r=32 use compared to the OLMo-7B model generating a chain-of-thought response?
2. How sensitive is the final model's performance to the choice of `k` (Truncated Backpropagation)?

**Ethical Concerns:**

["NO or VERY MINOR ethics concerns only"]

**Final Justification:**

5: Accept: Technically solid paper, with high impact on at least one sub-area of AI or moderate-to-high impact on more than one area of AI, with good-to-excellent evaluation, resources, reproducibility, and no unaddressed ethical considerations.

**Limitations:**

See weakness and questions.

**Quality:**

3

**Strengths And Weaknesses:**

Strengths:
1. The paper successfully trains a 3.5B parameter model and provides comprehensive benchmarks to show the method's effectiveness.
2. The architecture's ability to support adaptive compute, KV-cache sharing, and self-speculative decoding is interesting.
3. The visualization of token trajectories in latent space reveals emergent structured patterns like "orbits" and "sliders". This provides evidence for how the model performs abstract computations.

Weaknesses:
1. The authors are transparent that due to compute constraints, only one large-scale model was trained. While the results are promising, this makes it difficult to fully disentangle the effects of the architecture from the specific data mix and other hyperparameters.
2. Although positioned as an alternative to Chain-of-Thought, the paper lacks a direct performance comparison under an equivalent test-time FLOP budget. Such a comparison would be highly informative for evaluating the trade-offs between the two approaches.

---

> ### Author Rebuttal · Authors · 2025-07-31
>
> Thank you for your feedback and strong interest in our work. We’re glad that you find the model we trained interesting, and the adaptive compute contributions and visualizations meaningful. We’ll answer your questions below:
>
>
>
> ---
> &nbsp;
>
> > alternative to Chain-of-Thought, the paper lacks a direct performance comparison under an equivalent test-time FLOP budget
>
> We actually thought about this, but it was not so clear to us how to do a fair comparison to (verbalized/normal) CoT, as discussed also briefly in the limitations section. With our setup it is easy to control how much "latent CoT" happens by varying the number of recurrences $r$. Even on problems that do not require a lot of reasoning, we can still easily force a large value of r, and we do so in many parts of the paper.
>
> However, for models that should use verbal CoT, it is very hard to control the exact number of tokens that will be used for a given problem. A naive approach might involve simply suppressing truncation and/or forcing truncation and such an alteration to the natural generation process could have significant effects on output quality if CoT is cut short before the model would naturally chose to produce a final answer.
>
> As a concrete example, at the model scale we consider, we cannot really get the Olmo-7b models, or our non-recurrent baseline model to use chain of thought effectively, regardless of the desired output length. We considered prefilling these models with a CoT, but this would skew benchmark results to the model the CoT is extracted from. We can compare to CoT prompting on frontier open source models, such as Qwen 3, but those models are trained not only with 1-2 orders of magnitude more pretraining compute, but also extensively postrained to boost their CoT capabilities. It is unclear to us how to best elicit a clear comparison with verbalized CoT at the model scale we consider. We’d be happy for any pointers.
>
> ---
> &nbsp;
>
>
> > Could you provide a more direct comparison of the inference FLOPs
>
> Yes, we’ll add this as a column in e.g. Table 1. For now, you can consider Fig.1a (top x-axis) a ballpark estimate - although we note that actual inference costs are (in most scenarios) memory bound at generation time, which advantages the recurrent-depth model, which is using much less space to store its parameter, and less communication bandwidth between workers than a FLOP-equivalent fixed depth model (which is using every parameter only once per generated token). This difference is especially prominent when using cache sharing as described in section 5.
>
> ---
> &nbsp;
>
>
> > How sensitive is the final model's performance to the choice of k (Truncated Backpropagation)?
>
> We unfortunately cannot provide an answer at the 3.B scale. For the small-scale models (132M parameters trained for 10B tokens) we trained, we find that increasing $k$ starting from $k=1$ continuously improves the final perplexity when training over a fixed token budget, at the cost of additional memory, although the effects level off after $k=8$, with $k=16$ reaching the same perplexity. As such we chose to use $k=8$ for the pretraining run. We’d be happy to provide these small-scale ablations in the appendix.
>
> ---
> &nbsp;
>
>
> Overall, thank you for your feedback. We’re glad that you found our submission interesting! We hope findings and our “data point” of a large-scale recurrent model provide new information to the community that informs future efforts in this direction.
>
> Our goal with this work was to build upon conceptual findings on recurrence in transformers, and propose a scalable architecture and training setup that we could train as a large language model. We're pretty excited that the final model is indeed able to make use of more compute on reasoning-heavy tasks, and we hope to have accurately presented our findings and the limitations thereof in our submission.
>
> Please feel free to let us know if there are any follow-up comments or other questions that have come up since!

---

> > ### Comment · Reviewer_TR9E · 2025-08-04
> > **Thank you for your reply**
> >
> > Thank you for the response, authors. I have read the other reviewers' feedback as well as your responses to them. Please incorporate the additional insights and results you provided to the paper. Good luck!

---

### Official Review · Reviewer_DAK3 · 2025-07-03

**Clarity:** 4
**Significance:** 3
**Originality:** 4
**Rating:** 5
**Confidence:** 4

**Summary:**

This paper introduces a recurrent module in transformer blocks to allow for adaptive compute at inference time. The recurrent module is able to run for any number of iterations, and the main argument that the amount of inference compute can be controlled by using different number of iterations at test time. This work pretrained a 3.5B language model using the introduced recurrent module, and found that the resulting model gets different levels of performance when using different numbers of iterations in the recurrent module: using more iterations leads to higher inference performance. This work also analyzes the trained model to evaluate where recurrence helps and also finds potential mechanisms emerging for challenging questions.

**Questions:**

1. What's the benefit of training a recurrent module over increasing the number of layers of a transformer? If the goal is to achieve adaptive inference compute, can't you simply randomly drop transformer layers during training?
2. What is the fixed-depth baseline in table 4 exactly? What do you mean by a non-recurrent twin? Is that using a fixed number of iterations in the recurrent module?

**Ethical Concerns:**

["NO or VERY MINOR ethics concerns only"]

**Final Justification:**

Thanks authors for your response! I'm happy to keep my support for this paper, and I'll maintain my score.

**Limitations:**

Yes

**Paper Formatting Concerns:**

No issues found.

**Quality:**

3

**Strengths And Weaknesses:**

Strengths:
1. The idea of using recurrence to allow for adaptive compute at inference time is appealing.
2. Experiments pretrained a 3.5B model using the proposed module from scratch, which shows the potential of this approach on large models.

Weaknesses:
1. The potential benefit of using recurrence is to achieve latent reasoning, but experiments on math and reasoning benchmarks still rely on CoT reasoning which is not latent. Can you test on no CoT prompting (similar to implicit CoT works) on math and reasoning benchmarks to see if the recurrence can achieve latent reasoning without still having to rely on explicit CoT?
2. The median number of sampled iterations is 29, so at training time 50% of examples have seen recurrence for 29 or more iterations. However, looking at Figure 1, at inference time the performance has saturated at 20 iterations, way before 29 iterations. This is disappointing as the hope for the recurrent module is that by using more compute we can get higher performance; however, the results suggest that it doesn't generalize to anything beyond seen at training, and maybe it's better to simply fix the number of iterations to 20 and train a model.
3. The recurrence module used a Gaussian noise as input, but the effectiveness of doing so is not justified in experiments. At inference time, is this noise vector set to 0 or still being sampled from the Gaussian? Which version gets better performance?
4. While there are some mechanism analyses, it's still not clear what kind of reasoning the model is performing.

---

> ### Author Rebuttal · Authors · 2025-07-31
>
> Thank you for your detailed feedback and strong support of this submission. We’ll answer your questions below:
>
> > Can you test on no CoT prompting
>
> We incentivize the model to use (verbalized) CoT via prompting only when testing the “GSM8k CoT” benchmark task; all other evaluations do not use CoT prompting. That the model further improves when using both recurrent depth and verbal CoT prompting on GSM8k (both GSM8k versions can be compared in Table 2) is an interesting finding we wanted to highlight, connecting to our hypothesis in the introduction that both forms of reasoning could be complementary.
>
> ---
>
> > The median number of sampled iterations is 29, so at training time 50% of examples have seen recurrence for 29 or more iterations. However, looking at Figure 1, at inference time the performance has saturated at 20 iterations, way before 29 iterations.
>
> While we do want to point out that this behavior is highly task-specific, with e.g. reasoning tasks like GSM8k saturating later - we do note that the sigmoid-like benchmark accuracy behavior we observe is a finding of our work. We’ll extend our discussion of this finding in the next version. From the perspective of the algorithmic reasoning literature, this is not optimal behavior, and we would have preferred if increasing recurrence continued to increase performance on all tasks forever.
> However, from the modeling perspective, we believe that this observation should be considered in the context of the data the model is trained on and the relationship that data has to other forms of training for extended reasoning.
>
> ---
>
> 1. First, during pretraining, the model mostly observes general web-text. On webtext, most next tokens cannot actually be predicted with increased compute, and the best strategy is to memorize associations and patterns. Examples include the names of presidents, trivia facts, and many  basic constructs in natural language. Even for tokens that can be predicted with more compute the model must also learn to even identify this property; the fact that that more computation might be useful e.g. for a math question answer, but not for the name of the person answering the question, is a prediction problem in and of itself. We already consider it mildly surprising that the model has learned to use much more compute on tasks like GSM8k, where more compute might indeed be useful, simply from our pretraining mixture. We also wonder to what extent the accuracy on GSM8k is a function of compute for the model, or a function of understanding each query? For example, on the 55% of unsolved questions, does the model understand the query correctly but fail to use its recurrence to compute the correct answer, or does the model misunderstand the query and then spend an arbitrary amount of compute on “the wrong computation”? We hope to discern the difference between these cases in future work.
> 2. Second, the fact that extending the amount of reasoning beyond the training regime does not improve performance, is (maybe unfortunately) also a finding that comes up for verbalized CoT (as exemplified by deepseek-r1/openai o1 and successors). While the length of verbalized reasoning chains in these models increases during post-training for reasoning, after the model is posttrained in this manner, it cannot be prompted in a well defined way to reason “_even longer_” than what was observed during post-training. Moreover, recent results suggest, counterintuitively, that the length of a reasoning trace can be inversely proportional to the model’s expected accuracy on the problem.
>
> In summary, the observed sigmoid behavior may be attributable to a particular choice we made in pretraining the model, or it might be a more fundamental attribute. In our submission we have tried to neutrally and accurately depict this finding from the outset, such as in Fig 1 - but we can certainly extend some parts of the discussion in a similar manner as we did here.
>
> ---
>
> > The recurrence module used a Gaussian noise as input
>
> We did not have compute to ablate this choice at the 3.5b scale, but from our small-scale experiments, and from prior literature on path dependence, the random initialization of state proved helpful to stabilize the recurrence (we test that the final model is indeed path-independent in Fig.22-24 in the appendix). At test time, we also use the same random initialization on every token. We have ablated other choices at inference time, such as an initialization scale of 0, corresponding to your suggestion, (and higher, such as 2 or 10) and on some tasks there actually is a minor benefit to using a scale of 0 when using fewer iterations (e.g. 4 and 8 on GSM8k), but does not lead to gains when running r>16. We’ll provide this ablation as a figure in the next version of the submission.
>
> ---
>
> > what kind of reasoning the model is performing.
>
> We agree that this is an interesting question! Our preliminary study in Section 6 provides an initial attempt, but we hope that future work can help develop more concrete and interpretable descriptions of the type of latent reasoning that emerges inside this class of depth recurrent architectures.
>
> As one more examples of experiments we have done this submission, when decoding directly from latent state of earlier iterations, we do find that, e.g. on arithmetic questions, the latent state correlates most strongly with a certain number token for a part of its trajectory, before gradually decreasing the prominence of that token and correlating more strongly with another number token, such as the correct answer. When decoding into words early, this looks like the model is oscillating between two possible answers during the recurrence, but in latent space, both options are simultaneously evaluated and the model appears to collect evidence for the correct answer over time.
>
> ---
>
> > What's the benefit of training a recurrent module over increasing the number of layers of a transformer?
>
> While using a transformer with early exits is also a feasible strategy that we do not want to disparage, we do want to point to some advantages of recurrence:
> 1. Modeling adaptive compute with recurrence naturally leads to smaller models, which have practical advantages even beyond reasoning due to reduced communication costs during training and reduced storage requirements on consumer cards during inference.
> 2. For a transformer with early exits, computation and memory lookup are entangled, and an early exit might miss data stored in later layers, whereas for the recurrent model, the amount of data memorized in the recurrent block is fixed and only the amount of computation is varied.
> 3. Later training phases can modify the amount of recurrence that the recurrent depth model uses further, for example by gradually adapting the model to use more recurrence than during pretraining. This would not be possible for a transformer with early exits, where the maximal number of layers is predetermined. We have experimented with cooling down the model with a step distribution with a mean at 48, which does increase the performance of the model when evaluated with more recurrence (about +2% on GSM8k, and 4% on human-eval when evaluating r=64). We’ll add these additional findings to the next version.
>
> ---
>
> > What is the fixed-depth baseline in table 4 exactly? What do you mean by a non-recurrent twin?
>
> We’re sorry for compressing the legend of table 4 so heavily, this will be fixed in the updated version. The fixed-depth non-recurrent twin model is a 2nd model initialized with the same model architecture and number of parameters, trained on the same data in the same order, but without recurrence. Another way to describe it is that this is the recurrent-depth model, but it always recurs exactly once. This model has the same knowledge and observed the same data, but cannot use recurrence to increase its compute.
>
> ---
>
> Overall, our goal with this work was to build upon conceptual findings on recurrence in transformers, and propose a scalable architecture and training setup that we could train as a large language model. We're pretty excited that the final model is indeed able to make use of more compute on reasoning-heavy tasks. This is a property that emerged naturally from pretraining, and it was not obvious to us a-prior that this would happen - we also considered the possibility of a negative result, for example, where all curves in Fig.1a would be parallel - but, we find that the model does use recurrence in a constructive manner (as summarized in Fig.1b). There certainly is room to train this model more efficiently in the future, or propose training schemes and objectives with different recurrence characteristics, and we hope to have taken a first step in this direction with this submission.
>
> In conclusion, thank you for your thoughtful feedback! We hope that for some of your questions, our recent thoughts over the summer as summarized above on how to situate this work and how to think about its findings both in retrospect and for future development have proven helpful. Let us know if you have any further feedback, or if there are any follow-up questions we might be able to answer.

---

### Official Review · Reviewer_7vy3 · 2025-07-06

**Clarity:** 3
**Significance:** 3
**Originality:** 3
**Rating:** 5
**Confidence:** 3

**Summary:**

- This paper proposes a novel language model architecture capable of scaling test-time computation by reasoning in latent space rather than through explicit token generation.

- The model iteratively applies a recurrent block, allowing it to unroll to arbitrary depth at inference time, in contrast to conventional chain-of-thought methods that increase compute by generating more tokens.

- Unlike prior reasoning models, this approach does not require specialized training data, can work with small context windows, and is able to model reasoning patterns not easily expressed in natural language.

- The authors train a 3.5B parameter model on 800B tokens as a proof of concept, demonstrating that reasoning performance improves with increased compute allocation at inference time, particularly on math and coding tasks.

- The architecture also supports adaptive per-token compute, KV-cache sharing, and speculative decoding to reduce overall inference costs.

**Questions:**

See Strengths And Weaknesses.

**Ethical Concerns:**

["NO or VERY MINOR ethics concerns only"]

**Limitations:**

See Strengths And Weaknesses.

**Paper Formatting Concerns:**

No issue.

**Quality:**

4

**Strengths And Weaknesses:**

- Strengths:
  - The description of the **scalable recurrent architecture in Section 3 is clear and well-articulated**. In particular, Lines 78–83 are elegantly written and convey the design motivation effectively. The details of the model configuration in Section 3.2 are also clearly presented.

  - The paper conducts **comprehensive baseline comparisons**. Table 1 benchmarks the proposed model against Pythia and other models with comparable parameter sizes, while Appendix Table 7 extends the comparisons to recent strong models such as Qwen2.5 and LLaMA 3, making the evaluation thorough and convincing.

  - **The analysis of the role of Recurrence is detailed and well-explained**. Notably, Section 5 provides clear discussions on its interaction with KV cache reuse, self-speculative decoding (SD), and other aspects, demonstrating the broader significance of Recurrence beyond just computational scaling.

- Weaknesses
  - It would be valuable to include comparisons with other **recent large reasoning models** of a similar scale, such as Qwen 3, to further contextualize the model’s performance.

  - The discussion on the role and benefits of **Recurrence in the main text is somewhat redundant in places**. **A better organization** and separation of detailed analyses between the main paper and the Appendix could improve clarity and readability.

---

> ### Author Rebuttal · Authors · 2025-07-31
>
> Thank you for your feedback! We are glad that you find the proposed approach well-articulated, the comparisons comprehensive, and the analysis of the role of recurrence interesting.
>
> ---
> &nbsp;
>
> Regarding your questions,
> > It would be valuable to include comparisons with other recent large reasoning models of a similar scale, such as Qwen 3, to further contextualize the model’s performance.
>
> We'd be happy to update Table 7 to include the newest models that have come out, such as Qwen3, and will do so for the next version of this work.
>
> ---
> &nbsp;
>
>
> > The discussion on the role and benefits of Recurrence in the main text is somewhat redundant in places
>
> Regarding the potential redundancy of some explanations, we have preferred to err on the side of caution and assume that not all readers will understand certain aspects of the novel architecture or training process if presented in only one way. Rather we reiterate certain concepts in order to “bring our readers along for the ride” throughout the manuscript. While we hope this improves the readability of the paper in general, we will be sure to go over the draft with a fine-toothed comb when preparing the camera ready version to catch any unintentional artifacts from the drafting process. If there are specific passages where you felt strongly that the redundancy was distracting, we’d be happy to take another look and rewrite them.
>
> ---
> &nbsp;
>
>
> Overall, thank you for your strong support for this submission and idea, and we’d be happy to respond to further questions.

---

### Decision · Program_Chairs · 2025-09-17

**Decision:**

Accept (spotlight)

**Comment:**

This paper proposes a recurrent/looped architecture that can “reason in latent space” for inference-time scaling. A prototype model with 3.5B parameters is pre-trained on 800B tokens with this architecture and is shown to scale at inference-time for math and coding benchmarks. Overall all reviewers were very positive about the paper. They appreciated the novelty, clarity of presentation, comprehensive experimentation and inclusion of additional features like kv-sharing and self-speculative decoding. There were questions about the motivation of such an architecture in comparison to CoT and whether it is worth training such a model if deeper models are better. The authors provided reasonable responses to these, while acknowledging that this approach may not be compute optimal or ideal for knowledge intensive tasks. Overall this is a solid paper that concretely demonstrates the benefits of looped architecture and latent reasoning, topics that have both recently garnered interest.